



# Snow cover variations across China from 1952-2012

Xiaodong Huang[1*], Changyu Liu[2], Yunlong Wang[2], Qisheng Feng[2], and Tiangang Liang[2*]

[1] School of Geographical Sciences, Nanjing University of Information Science and Technology, Nanjing 210044, China

[2] State Key Laboratory of Grassland Agro-ecosystems, College of Pastoral Agriculture Science and Technology, Lanzhou University, Lanzhou, 730020, China

*Correspondence to*: Xiaodong Huang (huangxd@lzu.edu.cn)

**Abstract.** Based on a snow depth (SD) dataset retrieved from meteorological stations, this experiment explored snow indices including SD, snow covered days (SCDs), and snow phenology variations in China from 1952 to 2012. The results indicated that the snow in China exhibits regional differences, and the snow cover is mainly concentrated in three snow cover areas in Northeast China, northern Xinjiang and the Tibetan Plateau. In China, the annual average SD showed an increasing trend, and the increases in the average snow depth ($SD_{average}$), cumulative snow depth ($SD_{overall}$) and maximum snow depth ($SD_{max}$) reached 0.04 cm, 0.05 cm and 0.07 cm per decade, respectively. The significant increases were mainly concentrated in areas higher than 40 °N latitude, especially in Northeast China. The $SD_{average}$, $SD_{overall}$ and $SD_{max}$ jump points are mainly in 1956, 1957, 1978, and 1987. In the first main period, the $SD_{overall}$ oscillation in China is relatively stable, and its average period is approximately 13 years. The SCDs showed an increasing trend, with an increase of 0.5 days per decade. The significant increases in SCDs were also concentrated in areas higher than 40 °N latitude, especially in Northeast China. However, in the Tibetan Plateau, the decrease in the SCDs reached 0.1 days per decade. In snow phenology, the snow duration days (SDDs) of China decreased, and 17.4% of the meteorological stations showed significant decreasing trends. This result is mainly caused by the postponement of the snow onset date (SOD) and the advancement of the snow end date (SED). Geographical factors, including latitude, longitude and altitude, affect snow cover distribution directly and indirectly. The squared multiple correlations of SDDs and SCDs are greater than 0.9. Among the effects of SDDs and SCDs, the largest standardized total effect is from altitude on the SDDs, and the effect reaches 0.8.

## 1 Introduction

Snow cover, an essential part of the climate system, is a requisite part of the process of energy exchange and hydrological cycle in the global climate system (Parajka et al., 2010; Zhang et al., 2018).Due to the snow cover contains unique physical and chemical properties, including the high albedo and the sensitivity of physical form to temperature (Xiao and Che, 2016). In terms of energy balance, the change of snow cover affects the long-wave radiation, the short-wave radiation, the latent heat and sensible heat(Sade et al., 2014), which will affect the radiation balance of the surface to a large extent by underlying surface (Euskirchen, 2007; Xiao and Che, 2016; Yu et al., 2016). Because the snow cover has higher reflectivity and lower thermal conductivity, it can be regarded as the insulation layer of land system in winter. Therefore, the snow cover plays a significant role in global radiation





balance. And this balance is a major driver of global atmospheric circulation and relevant climate
change (Govindasamy and Caldeira, 2000).

Snow covers 40% of the global land surface in winter, and more than 90% of the seasonal snow
cover is concentrated in the Northern Hemisphere (Hall et al., 1995; Armstrong and Brodzik, 2001),
covering an area of approximately $4.6 \times 10^7$ km$^2$ (Frei and Robinson, 1999). In the global hydrological
cycle, snow cover not only affects the water cycle path in many ways but also is a very crucial form of
water resource storage (Ambadan, 2017; Shams et al., 2018). However, snow can also have a negative
impact on human life because snowfall and meltwater are the direct causes of snowmelt erosion,
snowmelt floods, avalanches, glacier landslides and other natural disasters (Li and Simonovic, 2010;
Chen, 2016). Therefore, snow cover has both positive and negative effects on the natural ecology and
human life (Liang et al., 2004; Jacobson, 2014; Dudley et al., 2017).

The Intergovernmental Panel on Climate Change (IPCC) has reported that climate warming over
the past 50 years is indisputable and that the temperature over the past 50 years is likely to be the
highest on average over the past 500 years (IPCC, 2013). Wang et al. (2018) found that the snow cover
area in the Northern Hemisphere showed a decreasing trend by using the MODIS snow products from
2000 to 2015, in which the snow cover area in high-latitude and high-elevation mountainous regions
decreased significantly, while the snow cover area in some middle and low latitudes showed increasing
trends. Zhong et al. (2018) found that both the annual mean and the maximum snow depth showed
significant increasing trends over the entire Eurasian continent. At the same time, the variation in the
snow depth over Eurasia differed seasonally. The snow depth decreases in autumn and increases in
spring and winter. China's annual mean snow covered area accounts for 27% of the country's total area
in the winter (Huang et al., 2016). Ke et al. (2016) found that a delayed snow onset date (SOD) and
advanced snow end date (SED) are common in China during the period of 1952 to 2010. There are
three snow cover areas in China: Northeast China, northern Xinjiang and the Tibetan Plateau (Liu et al.,
2014). The Tibetan Plateau is the region with the highest elevation and the deepest SD at the middle
latitudes of the Northern Hemisphere (Ma et al., 2010). The variation in the range of snow cover over
the Tibetan Plateau exceeds that over other regions of the Northern Hemisphere at the same latitude
(Yao et al., 2013), and a large number of studies have found that the snow covered days (SCDs) on the
Tibetan Plateau show a declining trend (Yao et al., 2013; Chen et al., 2015; Huang et al., 2017; Qiao et
al., 2018).

In the context of global warming, the feedback of the snow cover in China on climate change is
still unknown. Snow indices are often calculated from meteorological station data, which have great
advantages in the process of long time series research (Zhong et al., 2018). In this study, we aim to
explore the snow cover variations in China from 1952 to 2012 based on a SD dataset retrieved from
meteorological stations. The objectives are to 1) evaluate the spatial distribution and changing trend of
snow cover indices across China, 2) ascertain the variation trends and fluctuation periods of snow cover
indices in China, 3) compare the trend of snow cover indices in the three stable snow cover areas of
China, and 4) explore the reasons for the snow cover distribution characteristics in China.

## 2   Data and methodology

### 2.1 Data



The daily SD data in China from 1 January 1951 to 31 December 2013 were used, which were
provided by the National Meteorological Information Center of the China Meteorological
Administration (CMA) (http://data.cma.cn/en). A hydrological year spanned from July 1 of the current
year to June 31 of the ensuing year. The station locations are shown in Fig. 1. Initially, the standards of
the data quality control in this study were as follows. 1) In this study, only daily SD values larger than
1 cm were recorded as snow cover; regions with values less than 1 cm were regarded as no snow. 2) To
ensure the reasonableness of statistical analysis, we must ensure that the station datasets used for
statistical analysis were longer than 10 years. Therefore, the stations with less than 10 years of records
were omitted from the analysis. 3) To ensure the integrity of the data recorded by the stations during a
hydrological year, the data from the first hydrological year were omitted from the analysis. The first SD
data recorded by the station cannot be used to determine whether it is the first snowfall in the
hydrological year. As an example, since the SD measurements began in October 1951, the data from
July to October that year were missing; thus, in the first hydrological year the data of each station is
defective.

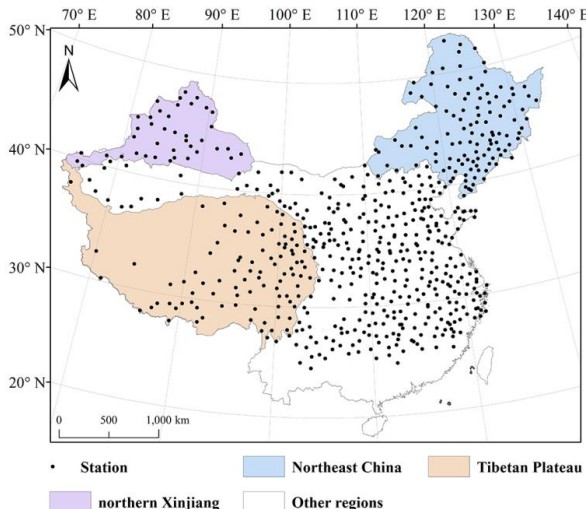

**Figure 1.** Geographical locations of the meteorological stations and the snow cover areas across China.

The snow cover indices, including the annual average snow depth ($SD_{average}$), cumulative snow
depth ($SD_{overall}$), maximum snow depth ($SD_{max}$), snow cover days (SCDs), snow onset date (SOD),
snow end date (SED), and snow duration days (SDDs), are calculated in this study. The $SD_{average}$ is
calculated by dividing the sum non-zero snow depth values by the total number of SCDs when ground
covered by snow in a hydrologic year (from 1st July to 30st June of the following year). The annual
$SD_{overall}$ is calculated by dividing the sum of snow depth records by the total number of days in the
hydrological year. The $SD_{max}$ is the maximum of the snow depth for the corresponding hydrologic year.
The SOD is calculated as the first date of snow onset in the hydrological year. The SED is regarded as
the snow end date in a hydrological years. The SDDs are the number of days from the SOD to the SED
in the corresponding hydrological year. And the SCDs is defined as the total days as snow covered
ground throughout the hydrological year. The difference between SCDs and SDDs is that SDDs





includes period of snow cover interruption during the snow season. The above snow indices are
calculated for each station during the corresponding hydrological years.

**2.2 Methodology**

a) Mann-Kendall test

Linear fitting is the most common and most extensive trend analysis method. Moreover, the
Mann-Kendall test (M-K) is also recommended by the World Meteorological Organization and is
widely used (Milan, 2013), which is frequently used to analyze the trends of changes in meteorological
and hydrological elements (Huang et al., 2016; Wang et al., 2018). In this study, two methods were
used to analyze the trends of the variations in snow cover elements from 1952 to 2012. The M-K

formulas are as follows:

$$S = \sum_{j=1}^{n-1} \sum_{i=j+1}^{n} sign(x_i - x_j) \qquad (1)$$

Where $n$ is the number of datasets to be analyzed. $x_i$ and the $x_j$ are the values in the time

series $i$ and $j$.

$$\begin{cases} Z = \dfrac{(S-1)}{\sqrt{\dfrac{n(n-1)(2n+5)}{18}}} & S > 0 \\ Z = 0 & S = 0 \quad (2) \\ Z = \dfrac{(S+1)}{\sqrt{\dfrac{n(n-1)(2n+5)}{18}}} & S < 0 \end{cases}$$

where $Z$ is the value used to judge whether the trend is increasing or increasing in trend analysis.

When the $Z$ is positive, the trend is increasing. Negative values of $Z$ represent decreasing trends. At the

same time, by comparing the absolute value of $Z$ with the standard value of $Z$, we can judge the

significance of the trend. In this study, significance levels of α=0.05 and α=0.01 were used. If the

absolute value of $Z$ is greater than $Z_{0.05}$ or $Z_{0.01}$, the trend is statistically significant or extremely

significant.

$$S_k = \sum_{i=1}^{k} r_i \, , r_i = \begin{cases} 1, x_i > x_j \\ 0, x_i \le x_j \end{cases}, (j = 1,2,\dots,i; k = 1,2,\dots,n)(3)$$

$$E[S_k] = \frac{k(k-1)}{4}(4)$$





$$var[S_k] = \frac{k(k-1)(2k-5)}{72} \quad 1 \le k \le n \tag{5}$$

$$UF_k = \frac{(S_k - E[S_k])}{\sqrt{var[S_k]}} \tag{6}$$

$$UB_k = -UF_k', UF_k' = UF_{n-k} \tag{7}$$

Where $UF$ is the standardization value of $S$. Moreover, $UF'$ is obtained by the inverse

sequence of $UF$.

UF and UB can roughly provide the jump points of the meteorological sequence. However, the

jump points of the meteorological sequence can be further judged by combining the M-K and moving t

tests. The formula for the moving t test is as follows:

$$t = \frac{(\bar{x}_1 - \bar{x}_2)}{s\sqrt{\frac{1}{n_1} + \frac{1}{n_2}}} \tag{8}$$

$$s = \sqrt{\frac{n_1 s_1^2 + n_2 s_2^2}{n_1 + n_2 - 2}} \tag{9}$$

When $t$ is greater than $t_{0.05}$, the year corresponding to t is a jump point.

b) Slope

In this study, the slope method is employed to analyze the snow cover variation trend (Liu et al., 2018).
The formula is as follows:

$$slope = \frac{n\sum_{i=1}^{n} i x_i - \sum_{i=1}^{n} i \sum_{i=1}^{n} x_i}{n\sum_{i=1}^{n} i^2 - (\sum_{i=1}^{n} i)^2} \tag{10}$$

Where $n$ is the number of datasets to be analyzed. $x_i$ is the value in the time series $i$.

c) Wavelet analysis

The periods of fluctuations are not immutable. With a change in the analytical scale, the period of the
fluctuations will change accordingly. In a multitemporal scale analysis, we can obtain the fluctuation
periods of variables over multiple time scales. In this study, the cmor wavelet in Molet analysis is
adopted (Liu et al., 2016). The formula is as follows:





$$cmor(x) = \frac{1}{\sqrt{\pi Fb}} e^{2i\pi Fcx} e^{-\frac{x^2}{Fb}} \quad (11)$$

Where **Fb** is the bandwidth coefficient and **Fc** is the central frequency of the wavelet.

d) Structural equation modeling

Structural equation modeling (SEM) is based on a variable covariance matrix and is a statistical method to analyze the relationships between variables in recent years (Bagozzi and Yi, 2012; Liu et al., 2016; Mi and Haeyoung, 2018). This method synthesizes a variety of statistical methods, including path

analysis, regression analysis, and factor analysis. In this study, SEM was used to analyze the coupling of snow cover factors and geographical factors to explore the reason for snow heterogeneity. The tool used for SEM in this study is IBM SPSS Amos 24. The analytical method in the equation is maximum likelihood.

**3     Result**

**3.1 SD**

The mean annual $SD_{average}$, $SD_{overall}$ and $SD_{max}$ gradually increase with increasing latitude and altitude from 1952 to 2012 in China (Fig. 2a, c and e). The largest mean $SD_{average}$ and $SD_{overall}$ are both in Northeast China, with values of 24.8 cm and 9.4 cm. The largest mean $SD_{max}$ appeared in the Tibetan Plateau, reaching 56.3 cm. In China, the mean $SD_{average}$, $SD_{overall}$ and $SD_{max}$ are 4.2 cm, 0.7 cm and 9.3

cm, respectively. In China, $SD_{average}$, $SD_{overall}$ and $SD_{max}$ all showed increasing trends from 1952 to 2012 (Fig. 3a, c and e), and their increases were 0.05 cm, 0.04 cm and 0.07 cm per decade, respectively. The distributions of these trends were similar (Fig. 2b, d and f). The stations with significant increases in $SD_{average}$, $SD_{overall}$ and $SD_{max}$ were mainly concentrated at high latitudes in China. The proportions of meteorological stations with significant increases in $SD_{average}$, $SD_{overall}$ and $SD_{max}$ were 9.8%, 10.0% and

5.6%, respectively. The stations with significant decreases in $SD_{average}$, $SD_{overall}$ and $SD_{max}$ are mainly concentrated in central China. The proportions of meteorological stations with significant decreases in $SD_{average}$, $SD_{overall}$ and $SD_{max}$ are 6.8%, 5.0% and 4.2%. In the three snow cover areas of Northeast China, northern Xinjiang and the Tibetan Plateau, the trends of $SD_{average}$, $SD_{overall}$ and $SD_{max}$ are highly consistent with the overall trends in China (Table 1). In Northeast China, these increasing trends were

0.4 cm, 0.2 cm and 0.6 cm per decade. In northern Xinjiang, these increasing trends were 0.2 cm, 0.1 cm and 0.3 cm per decade. In the Tibetan Plateau, these increasing trends were 0.01 cm, 0.01 cm and 0.02 cm per decade.

**Table 1.** Trends in SD across three snow areas from 1952 to 2012 (**).

| Zone | Variate | Slope analysis | | M-K analysis |
|---|---|---|---|---|
| | | Slope | *P*-value | Z-value |
| Northeast China | $SD_{average}$ | 0.04 | 0.00** | 2.87** |
| | $SD_{overall}$ | 0.02 | 0.00** | 2.63** |
| | $SD_{max}$ | 0.06 | 0.01* | 2.03* |
| Northern Xinjiang | $SD_{average}$ | 0.02 | 0.14 | 1.36 |
| | $SD_{overall}$ | 0.01 | 0.19 | 1.03 |



| | | | | |
|---|---|---|---|---|
| | SD$_{max}$ | 0.03 | 0.24 | 1.36 |
| Tibetan Plateau | SD$_{average}$ | 0.00 | 0.73 | 0.35 |
| | SD$_{overall}$ | 0.00 | 0.33 | 0.81 |
| | SD$_{max}$ | 0.00 | 0.84 | 0.48 |

\* and \*\* denote text significant < 0.05 and < 0.01, respectively.

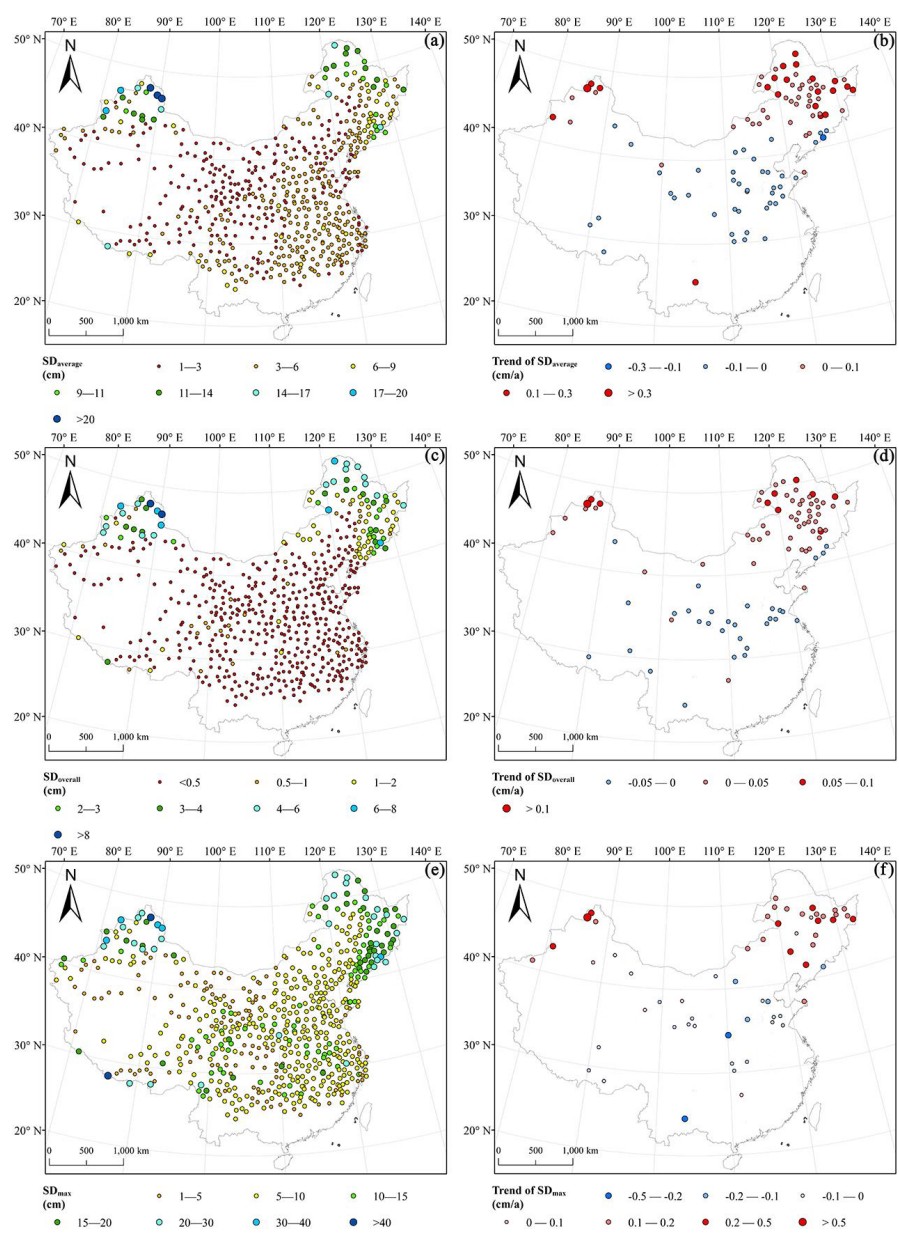

**Figure 2.** Panels (a), (c) and (e) represent the spatial distributions of the mean annual $SD_{average}$, the mean annual $SD_{overall}$ and the mean $SD_{max}$ across China, respectively. Panels (b), (d) and (f) show the distribution of the trends of the mean annual $SD_{average}$, the mean annual $SD_{overall}$ and the mean $SD_{max}$, respectively, as determined by the trend analysis to exhibit significant changes ($P< 0.05$) across China.

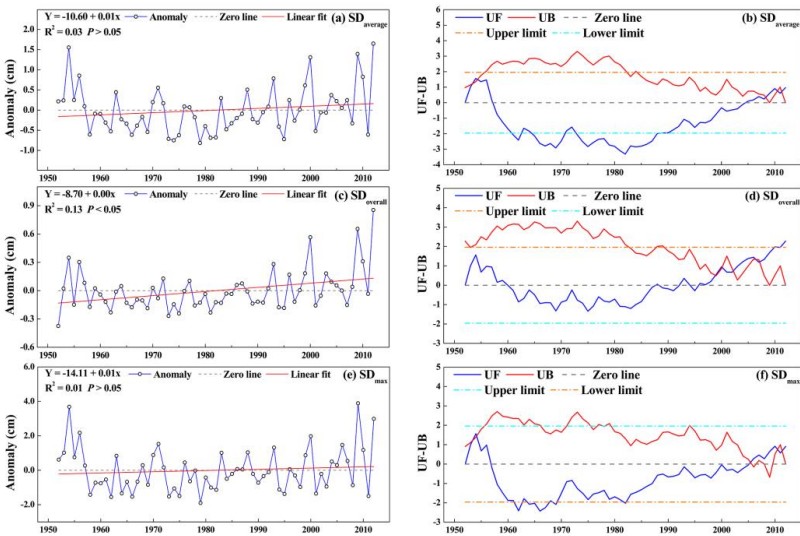


**Figure 3.** Panels (a), (c) and (e) represent the linear fit of the mean annual $SD_{average}$, the mean annual $SD_{overall}$ and the mean $SD_{max}$ in China, respectively. Panels (b), (d) and (f) represent the results of the M-K test of the mean annual $SD_{average}$, the mean annual $SD_{overall}$ and the mean $SD_{max}$ in China, respectively.

The results of the M-K trend test are the same as the results of the slope method (Fig. 3b, d and f). In the M-K test, when the UF is greater than 0, there is an increasing trend from the initial year to the corresponding year. When the UF is less than 0, there is a decreasing trend. In China, the overall trends of $SD_{average}$, $SD_{overall}$ and $SD_{max}$ first increase, then decrease, and finally increase again during the period of 1952 to 2012. The trend of $SD_{average}$ changes in 1958 and 2005. The trend of $SD_{overall}$ changes in 1960

and 1996. The changes in the $SD_{max}$ trend occur in 1957 and 2006.When the M-K and moving t tests were combined, the jump points of $SD_{average}$, $SD_{overall}$ and $SD_{max}$ could be identified (Table 2). In China, the jump points of $SD_{average}$ are in 1956, 1957 and 1978. For $SD_{overall}$, the jump point is in 1987. For $SD_{max}$, the jump points appear in 1956 and 1957. In Northeast China, the $SD_{average}$ jump point is in 1972. However, there is no significant jump point in $SD_{overall}$. The $SD_{max}$ jump points are in 1956 and 1957. In

northern Xinjiang, there are four identical jump points for $SD_{average}$, $SD_{overall}$ and $SD_{max}$, which appear in 1959, 1960, 1979, and 1987. In the Tibetan Plateau, the $SD_{average}$ jump points are in 1955 and 1956. There are two identical jump points of $SD_{overall}$ and $SD_{max}$, which appear in 1956 and 1997.

**Table 2.** The jump points of snow cover indices detected by a moving t test.

|  | $SD_{average}$ | $SD_{overall}$ | $SD_{max}$ | SCDs | SOD | SED | SDDs |
|---|---|---|---|---|---|---|---|
| China | 1956, 1957, 1978 | 1987 | 1956, 1957 | 1987 | 1958, 1999 | — | 1987, 2004 |





| Northeast China | 1972 | — | 1957, 1972 | 1972, 1987 | 1970, 2008 | — | 1957, 1987 |
|---|---|---|---|---|---|---|---|
| Northern Xinjiang | 1959, 1960, 1979, 1987 | 1959, 1960, 1979, 1987 | 1959, 1960, 1979, 1987 | 1987, 1988 | 1992 | 1958, 1959 | 1987 |
| Tibetan Plateau | 1955, 1956 | 1956, 1997 | 1956, 1997 | 1997 | — | 2005 | — |

In China, the SD$_{average}$, SD$_{overall}$ and SD$_{max}$ are related to geographical zonality. The distributions of
these values are concentrated in the three snow areas, including Northeast China, northern Xinjiang and
the Tibetan Plateau. The national SD$_{average}$, SD$_{overall}$ and SD$_{max}$ showed increasing trends from 1952 to
2012 under the context of global warming. In the three snow areas, the trends are highly consistent with
the overall trend in China, and the largest increase occurs in Northeast China, followed by northern
Xinjiang and finally the Tibetan Plateau.

**3.2 SCDs**

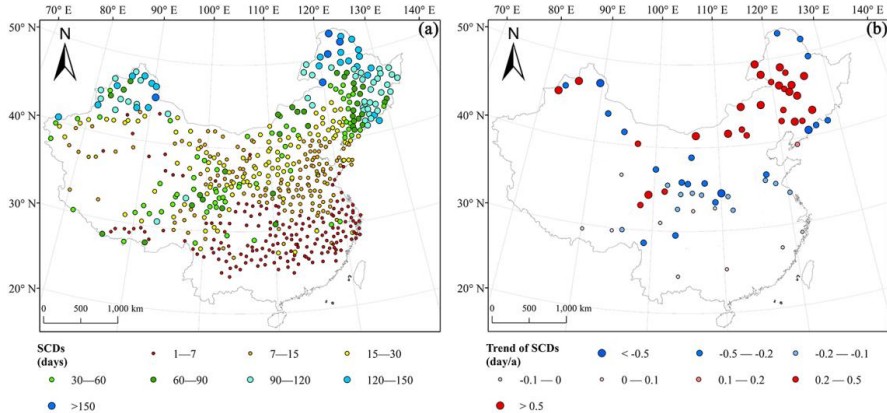

**Figure 4.** Panel (a) represents the spatial distribution of the mean annual SCDs. Panel (b) is the
distribution of the mean annual SCDs trend with significant changes (*P*< 0.05) as determined by the
trend analysis.

The mean annual SCDs gradually increase with increasing latitude and altitude from 1952 to 2012 in
China (Fig. 4a), and the mean SCDs is 34 days per year. The largest mean SCDs occurs in Northeast
China. The value reaches 168 days. In China, the trend in the number of SCDs increases from 1952 to
2012 (Fig. 5a) at a rate of 0.5 days per decade. The significant increases in the SCDs are mainly
concentrated in western Northeast China (110°E - 128°E, 40°N - 50°N) from 1952 to 2012 (Fig. 4b).
The proportion of meteorological stations with significant increases in the SCDs is 5.1%. The stations
with significant decreases are mainly distributed in central China. The proportion of meteorological
stations with significant decreases in SCDs is 6.5%. In the three snow cover areas (Table 3), the trends
of the SCDs in Northeast China and northern Xinjiang both increase. Especially in Northeast China, the
increase in the SCDs is significant. However, in the Tibetan Plateau, the trend of the SCDs is
decreasing. The trends of the SCDs in Northeast China, northern Xinjiang and the Tibetan Plateau are
2.4 days, 0.6 days and -0.1 days per decade, respectively.

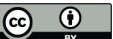



**Figure 5.** Panel (a) is the linear fit of the mean annual SCDs in China. Panel (b) represents the M-K of the mean annual SCDs in China.

**Table 3.** Trends in SCDs across three snow areas from 1952 to 2012.

| Zone | Slope analysis | | M-K analysis |
|---|---|---|---|
| | Slope | *P*-value | Z-value |
| Northeast China | 0.24 | 0.03* | 1.99* |
| Northern Xinjiang | 0.06 | 0.37 | 1.13 |
| Tibetan Plateau | -0.01 | 0.88 | -0.16 |

* denote test significant < 0.05

The results of the M-K test are the same as the results of the slope method. In China, the trend of SCDs first increases, then decreases and finally increases (Fig. 5b). The trend transformation occurs in 1964 and 1984. There were no significant trends in SCDs throughout the study period. When the M-K 235 and moving t test are combined (Table 2), the jump point of the SCDs in China occurs in 1987. In Northeast China, the jump points of the SCDs occur in 1972 and 1987. In northern Xinjiang, the jump points occur in 1987 and 1988. In the Tibetan Plateau, the jump point occurs in 1997. The distribution of SCDs exhibits regional differences. In China, the SCDs increase from 1952 to 2012; however, the SCDs decrease in the Tibetan Plateau in the context of global warming. The largest change occurs in 240 Northeast China, followed by northern Xinjiang and finally the Tibetan Plateau.

**3.3 Snow phenology**

The long SDDs frequently occur concurrently with earlier SODs and later SEDs (Fig. 6a, c and d). In China, the SDDs increase, the SODs advance, and the SEDs delay with increasing latitude and altitude from 1952 to 2012, and the mean SDDs, SODs and SEDs are 100 days, 157 days and 256 days, 245 respectively. The longest SDDs, the earliest SOD, and the latest SED all occur in the Tibetan Plateau, with values of 326 days, 29 days and 354 days, respectively. In China, the SDDs show a decreasing trend, which is reflected by the postponement of the SOD and the advance of the SED (Fig. 7a, c and e), and the trends of SDDs, SOD and SED are -0.7 days, 0.03 days and -0.6 days, respectively. The stations with significant trends in SDDs mainly show a trend of significant shortening across China 250 (Fig. 6b). The proportion of meteorological stations with significant shortening in SDDs is 17.4%. The proportion of meteorological stations with SDDs with significantly increasing trends is 0.3%. The stations with significant delays in SOD are mainly concentrated in Northeast China, Northwest China and the Tibetan Plateau (Fig. 6d), and the proportion of total meteorological stations is 9.5%. The stations with significantly advancing SODs are mainly concentrated in Southeast China, and the 255 proportion is 4.3%. The stations with significant trends in SED mainly represent the advancing trend (Fig. 6f), which represent a proportion of 13.0%, while the proportion of stations with significantly delayed SEDs is 0.3%.In the three snow cover areas, the trends of SDDs, SOD and SED are highly





consistent with the overall trends in China (Table 4). In Northeast China, northern Xinjiang and the Tibetan Plateau, the reductions in SDDs are 1.0 days, 1.0 days and 3.5 days per decade, respectively.

The postponement of SOD is 0.1 days, 0.6 days and 3 days per decade, and the advances of SED are 0.8 days, 0.3 days, and 0.5 days per decade.

**Table 4.** Trends in snow phenology across three snow areas from 1952 to 2012.

| Zone | Variate | Slope analysis | | M-K analysis |
|---|---|---|---|---|
| | | Slope | *P*-value | Z-value |
| Northeast China | SOD | 0.01 | 0.81 | 0.18 |
| | SED | -0.08 | 0.16 | -1.6 |
| | SDDs | -0.10 | 0.25 | -1.25 |
| Northern Xinjiang | SOD | 0.06 | 0.31 | 1.29 |
| | SED | -0.03 | 0.56 | -1.04 |
| | SDDs | -0.10 | 0.26 | -1.36 |
| Tibetan Plateau | SOD | 0.30 | 0.00** | 4.20** |
| | SED | -0.05 | 0.30 | -1.09 |
| | SDDs | -0.35 | 0.00** | -3.55** |



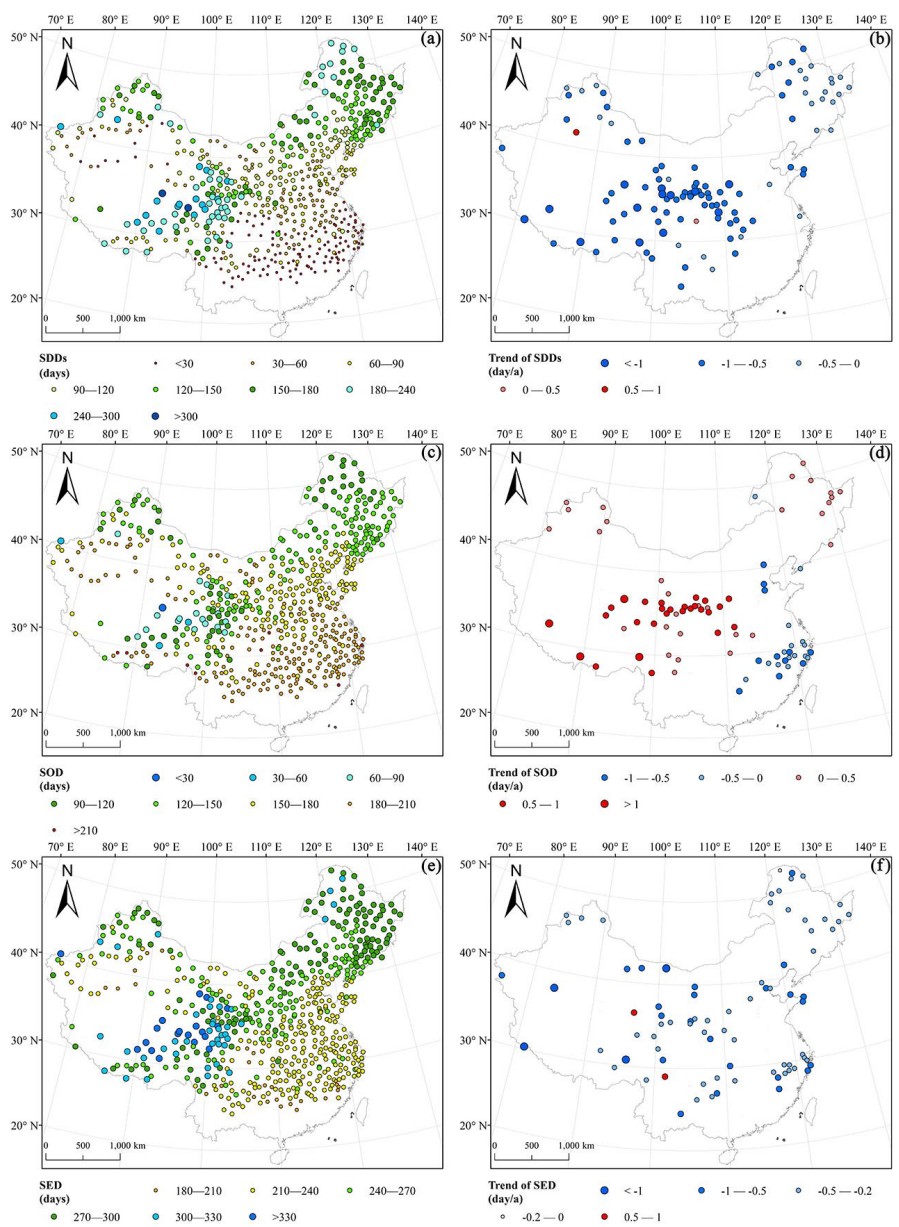

**Figure 6.** Panels (a), (c) and (e) represent the spatial distributions of the mean annual SDDs, the mean annual SOD and the mean SED across China, respectively. Panels (b), (d) and (f) are the distribution of the trend of the mean annual SDDs, the mean annual SOD and the mean SED with significant changes ($P < 0.05$) across China as determined by the trend analysis.

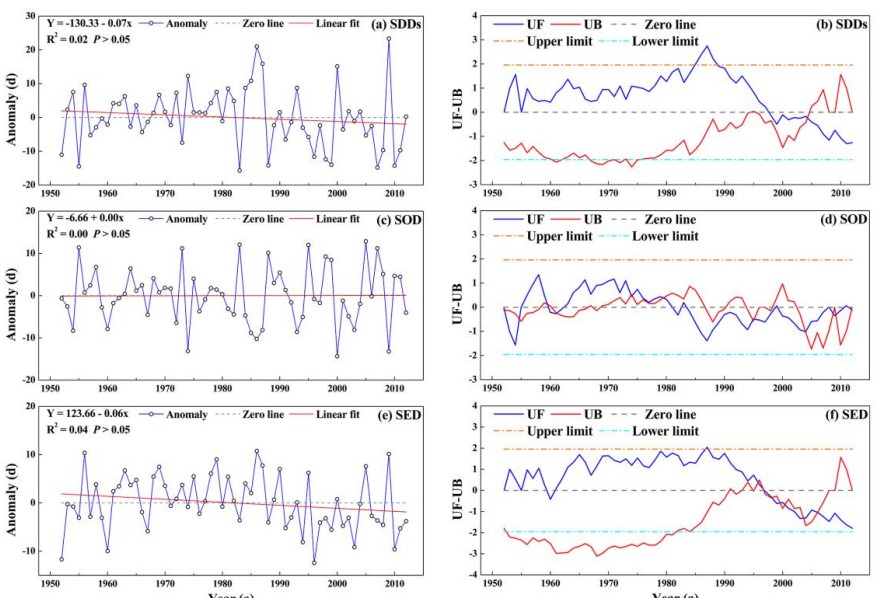

**Figure 7.** Panels (a), (c) and (e) represent the linear fit of the mean annual SDDs, the mean annual SOD and the mean SED in China, respectively. Panels (b), (d) and (f) represent the M-K test of the mean annual SDDs, the mean annual SOD and the mean SED in China, respectively.

The results of the M-K trend test are the same as the results of the slope method. In China, the trend of SDDs first increases and then decreases (Fig. 7b). The trend transformation occurs in 1998. The significant increases occur from 1985 to 1988. The trend of the SOD is first delayed and then advances (Fig. 7d). The trend transformation occurs in 1981. The trend of the SED is delayed at first and then advances (Fig. 7f). The trend transformation occurs in 1997. The M-K and moving t test are combined (Table 2), which indicates that the jump points of SDDs occur in 1987 and 2004. The SOD jump points occur in 1958 and 1999. There are no significant jump points for the SED in China. In Northeast China, the SDD jump points occur in 1957 and 1987. The SOD jump points occur in 1970 and 2008. There is no significant SED jump point in Northeast China. In northern Xinjiang, the SDDs jump point is in 1987. The SOD jump point occurs in 1992. The SED jump points occur in 1958 and 1959. In the Tibetan Plateau, there are no significant SOD and SDDs jump points. The SED jump point occurs in 2005. In China, the distributions of SDDs, SOD and SED are related to geographical zonality, and their trends are shortening, delaying and advancing from 1952 to 2012 under the context of global warming. Especially in the Tibetan Plateau, the changes in these trends are greater than those in other places.

### 3.4 Wavelet analysis

The Morlet wavelet is often used for wavelet analysis in atmospheric science research. The Morlet wavelet is a kind of multiresolution wavelet analysis with time and frequency properties, which provides a possibility for improved study of time series problems. The Morlet wavelet can clearly reveal a variety of change periods hidden in a time series.



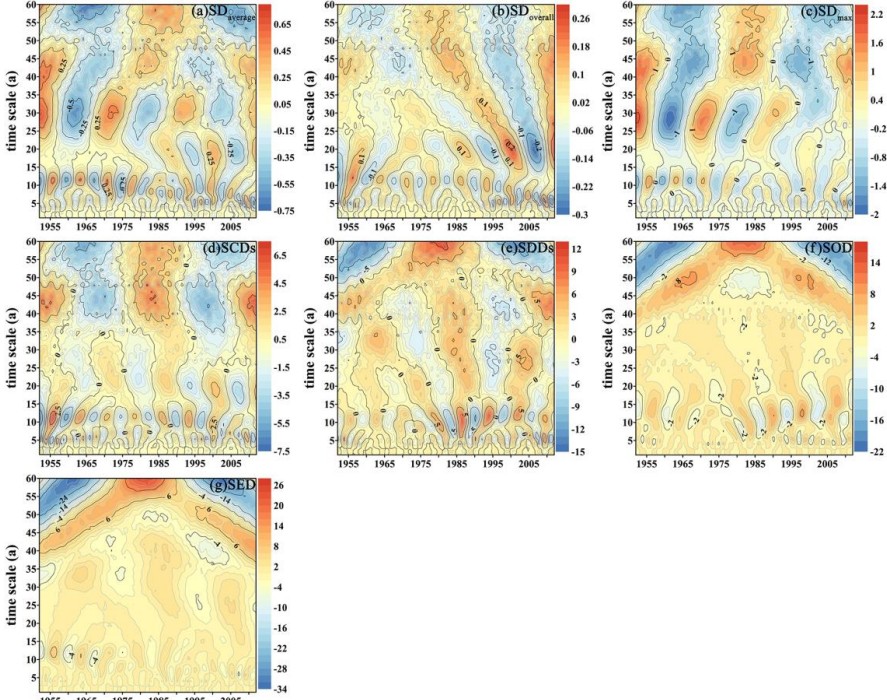

**Figure8.** Panels (a) to (g) show the contour maps of the real part of the wavelet coefficients for the
SD$_{average}$, SD$_{overall}$, SD$_{max}$, SCDs, SDDs, SOD, and SED, respectively.

In this study, the contour maps of the real part of the wavelet coefficients can reflect the periodic
changes in snow cover sequence at different time scales (Fig. 8). In Fig. 8, the abscissa is the year, the
ordinate is the time scale and the isopleth in the figure is the real part of the wavelet coefficient. When
the real part of the wavelet coefficient is positive, the index of snow cover is in the period that is more
than the mean. When the real part of the wavelet coefficient is negative, the index of snow cover is in
the period that is less than the mean. In Morlet wavelet analysis, the most oscillatory time scale is the
first main period of wavelet analysis. A great deal of research has focused on the oscillation of the first
main period (Liu et al., 2016). Therefore, the periodic oscillation of snow cover indices under the first
main period is explored.

The variations in SD$_{average}$, SD$_{overall}$, SD$_{max}$, SCDs, SDDs, SOD, and SED have unique
characteristics at multiple time scales (Fig. 8). However, the distributions of the oscillations in SDDs,
SOD and SED are similar. The results of the first main period are shown in Table 5. The first main
periods of SD$_{average}$, SD$_{overall}$, SD$_{max}$ and SCDs are 60 years, 20 years, 44 days and 60 days. The average
cycles are 35 years, 13 years, 30 years and 35 years. There are approximately two periods in SD$_{average}$,
SD$_{max}$ and SCDs throughout the study period. There are approximately five periods in SD$_{overall}$ from
1952 to 2012. The maximum centers of SD$_{average}$ are in 1952 and 1988, and the minimum centers of
SD$_{average}$ are in 1966 and 2003. The maximum centers of SD$_{overall}$ are 1959, 1973, 1987, 2000 and 2012,
and the minimum centers of SD$_{overall}$ are 1953, 1965, 1980, 1993 and 2006. The maximum centers of
SD$_{max}$ are in1953, 1982 and 2011, and the minimum centers of SD$_{max}$ are in 1969 and 1997. The





maximum centers of SCDs are in 1952 and 1985, and the minimum centers of SCDs are in 1963 and 2003.The first main periods of SDDs, SOD and SED are all 60 years. The average cycles are 40 years. There are approximately one and a half periods in SDDs, SOD and SED. The maximum center of the SDDs is in 1978, and the minimum centers of the SDDs are in 1959 and 2003.The maximum centers of SOD and SED both occurred in 1981. The minimum centers of SOD are in 1962 and 2002.The
minimum centers of SED are in 1962 and 2003.

**Table 5.** Results of wavelet analysis according to the first analysis scale.

| Variate | Analysis scale | Period | Maximum center | Minimum center |
|---|---|---|---|---|
| SD | 60 | 35 | 1952, 1988 | 1966, 2003 |
| $SD_{overall}$ | 20 | 13 | 1959, 1973, 1987, 2000, 2012 | 1953, 1965, 1980, 1993, 2006 |
| $SD_{max}$ | 44 | 30 | 1953, 1982, 2011 | 1969, 1997 |
| SCDs | 60 | 35 | 1952, 1985 | 1963, 2003 |
| SDDs | 60 | 40 | 1978 | 1959, 2003 |
| SOD | 60 | 40 | 1981 | 1962, 2002 |
| SED | 60 | 40 | 1981 | 1962, 2003 |

### 3.5 Snow heterogeneity

In the structural equation model, there are two types of variables, snow cover factors and geographical factors. Different datasets, including China, Northeast China, northern Xinjiang, and the Tibetan
Plateau, are used to run under the same model path, and the model fit is different (Table 6). Only the Northeast China dataset best matches the model because there is no significant difference between the model and the dataset ($P > 0.05$). Moreover, the indices of the model fit are close to the indices of the saturation model, including RMSEA GFI, CFI, NFI, and AIC. Therefore, in this study, the relationship between geographical factors and snow cover factors China is analyzed further in only Northeast (Fig.
9). In this model, all standardized regression weights reached a statistically significant level ($P<0.05$).

**Table 6.** The model fit parameters in SEM using a dataset of individual snow areas.

| Zone | CMIN/DF | Probability-level | GFI | RMSEA | CFI | NFI | AIC |
|---|---|---|---|---|---|---|---|
| Across China | 58.368 | 0.000 | 0.926 | 0.309 | 0.963 | 0.963 | 297.471 |
| Northeast China | 1.729 | 0.140 | 0.986 | 0.079 | 0.998 | 0.996 | 70.918 |
| Northern Xinjiang | 6.441 | 0.000 | 0.887 | 0.374 | 0.959 | 0.954 | 89.763 |
| Tibetan Plateau | 11.978 | 0.000 | 0.903 | 0.347 | 0.948 | 0.945 | 111.911 |

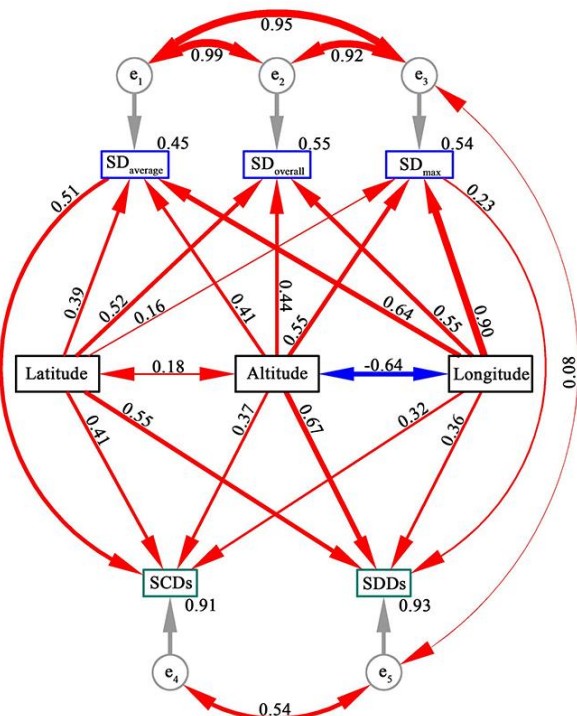

**Figure9.** The analysis of the influence of geographical factors on snow cover in Northeast China and the positive feedback effect of snow cover on itself. The thickness of the line increases with the increase in the effect.

In this model, the estimates of the squared multiple correlations of $SD_{average}$, $SD_{overall}$, MSD, SCDs, and SDDs are 0.5, 0.6, 0.5, 0.9, and 0.9, respectively. Therefore, SCDs and SDDs are more fully explained. To further explore the reason for the snow heterogeneity, two processes are employed. One process is the feedback of snow cover, and the other is the influence of geographical factors (Table 7). There is positive feedback in the snow cover to itself. SCDs increase with increasing $SD_{average}$. The standardized effect of $SD_{average}$ to SCDs is 0.5. SDDs increase with increasing $SD_{max}$. The standardized effect of $SD_{max}$ on SDDs is 0.2. Geographical factors have a positive impact on snow cover factors, which directly affect $SD_{average}$, $SD_{overall}$ and $SD_{max}$. The standardization effects of longitude on $SD_{average}$, $SD_{overall}$ and $SD_{max}$ are 0.6, 0.6 and 0.9. The standardization effects of altitude on $SD_{average}$, $SD_{overall}$ and $SD_{max}$ are 0.4, 0.4 and 0.6. The standardized effects of latitude on $SD_{average}$, $SD_{overall}$ and $SD_{max}$ are 0.4, 0.5 and 0.2. Due to the feedback of snow cover, geographical factors affect SCDs and SDDs directly and indirectly. The standardized total effects of longitude, altitude and latitude on SCDs are 0.7, 0.6 and 0.6. The standardized total effects of these factors on SDDs are 0.6, 0.8 and 0.6. $SD_{average}$, $SD_{overall}$, $SD_{max}$, SCDs and SDDs are not adequately explained by geographical factors. There is an extremely significant correlation between the parts that are not fully explained. Among $SD_{average}$, $SD_{overall}$, and $SD_{max}$, the standardized correlation coefficient of the parts that are not fully explained are as high as 0.9 or above. The standardized correlation coefficient of that parts between SCDs and SDDs is 0.6, and that between SDDs and SDmax is 0.1.



**Table 7.** Standardized total effects in SEM using a dataset from Northeast China.

|  | Longitude | Altitude | Latitude | $SD_{average}$ | $SD_{max}$ |
|---|---|---|---|---|---|
| $SD_{average}$ | 0.64 | 0.41 | 0.39 | — | — |
| $SD_{overall}$ | 0.55 | 0.44 | 0.52 | — | — |
| $SD_{max}$ | 0.90 | 0.55 | 0.16 | — | — |
| SCDs | 0.65 | 0.58 | 0.60 | 0.51 | — |
| SDDs | 0.57 | 0.79 | 0.59 | — | 0.23 |

## 4    Discussion

In recent years, the change in snow cover over China has attracted much attention. Using passive microwave remote sensing, Che et al. (2008) found that the SD in China showed a weak increasing trend from 1978 to 2006. The results of our study are roughly the same as those of this previous study. However, our study found that the areas where $SD_{average}$, $SD_{overall}$ and $SD_{max}$ increased significantly were mainly concentrated in latitudes above 40 °N. The results of both the M-K and slope methods show that the changes in the three snow areas are the same as the overall changes in China. The largest upward trends in $SD_{average}$, $SD_{overall}$ and $SD_{max}$ occur in Northeast China, followed by northern Xinjiang and the Tibetan Plateau.

Wang et al. (2018) found that the SCDs in the middle and low latitudes of the Northern Hemisphere, including Northeast China, showed an increasing trend from 2000 to 2015. However, our study obtains similar results by using a dataset from meteorological stations. In our study, we found that the SCDs increased in Northeast China, northern Xinjiang and even throughout China. However, the SCDs of the Tibetan Plateau are shortening. This result is similar to the results obtained by Huang et al. (2017), who used MODIS daily snow cover products from 2001 to 2014. Ke et al. (2016) found that from 1952 to 2010, the overall snow phenology in China showed a delay in the SOD and advancement of the SED. This result is similar to the results of our study. The snow phenology showed a shortening of the SDDs, a delay in the SOD and advancement of the SED in China. In the Tibetan Plateau, the trends of SOD and SDDs are significant. The trend of SOD (0.3 d/a) is much larger than that of SED (-0.1 d/a). Therefore, the main reason for the shortening of the SDDs in the Tibetan Plateau may be the delay of the SOD.

Due to the unique physical characteristics of snow cover, snow cover has a positive feedback effect to itself (Tedesco and Miller, 2007). As a very special underlying surface, snow cover affects the surface radiation balance to a large extent. With the increase in SD, the albedo is directly affected, which has a great influence on the temperature (Warren, 1982; Xue, 2017). At the same time, a large number of studies have proven that temperature is closely related to snow cover (Zhao, 2010; Qin, 2018; Ye, 2018). Nevertheless, few studies have provided data support for the positive feedback effect of snow cover. In our study, a structural equation model is used to provide more scientific and reasonable evidence that verifies the positive feedback effect of snow cover and the indirect effect of geographical factors on the temporal dimension of snow cover. In particular, SCDs and SDDs have the highest degree of interpretation. The squared multiple correlations of SCDs and SDDs are greater than 0.9.

## 5    Conclusion

The variation and distribution of snow cover and its causes have always been a popular topic. In this study, we use regression analysis, M-K analysis, wavelet analysis and SEM to analyze a snow depth



dataset from 1952 to 2012 across China, which is from the CMA. The main conclusions are condensed as follows:

(1) Snow cover in China is mainly distributed in Northeast China, northern Xinjiang and the Tibetan Plateau. The largest $SD_{average}$ and $SD_{overall}$ are in northern Xinjiang, with values of 24.8 cm and 9.4 cm,
respectively. The highest SCDs are in Northeast China, reaching 168 days. The largest $SD_{max}$, the longest SDDs, the earliest SOD, and the latest SED are in the Tibetan Plateau, with values of 56.3 cm, 326 days, 29 days, and 354 days.

(2) The overall trend of snow cover in China is that the trends of $SD_{average}$, $SD_{overall}$, $SD_{max}$, and SCDs are increasing and the trend of SDDs is shortening, which is caused by the delay of the SOD and the
advance of the SED from 1952 to 2012. The oscillation periods and extreme value centers of $SD_{average}$, $SD_{overall}$, $SD_{max}$, SCDs, SDDs, SOD and SED are different, while the oscillation periods among SDDs, SOD and SED are similar.

(3) Among the three snow cover areas, only the SCDs in the Tibetan Plateau are different from the overall trends in China, which is reflected by the decreasing trend of SCDs in the Tibetan Plateau.
Other indicators of the three snow cover areas are highly consistent with the trends in China. $SD_{average}$, $SD_{overall}$, $SD_{max}$, and SCDs are significantly increasing in Northeast China. The SDDs and SOD in the Tibetan Plateau are significantly shortening and delaying.

(4) There is some overlap in the $SD_{average}$, $SD_{overall}$, and $SD_{max}$ jump points. This phenomenon is most obvious in northern Xinjiang. In northern Xinjiang, there are four identical jump points in 1959, 1960,
1979, and 1987.

(5) Geographical factors have significant direct and indirect impacts on snow cover. The squared multiple correlations of SDDs and SCDs are greater than 0.9. The squared multiple correlations of $SD_{average}$, $SD_{overall}$, and $SD_{max}$ are approximately 0.5. The largest standardized effect on $SD_{average}$, $SD_{overall}$, $SD_{max}$, and SCDs is from longitude, and their effects are 0.6, 0.6, 0.9, and 0.7. While the
largest standardized effect on SDDs is from altitude, which reaches 0.8. The indirect effect is caused by the influence from $SD_{average}$ on the SCDs and from $SD_{max}$ on the SDDs, and their values reach 0.5 and 0.2.

**Acknowledgments**

This work was supported by the Natural Science Foundation Projects of China (41971293; 41671330),
the Science and Technology Basic Resource Investigation Program of China (2017FY100501), and the Startup Foundation for Introducing Talent of Nanjing University of Information Science & Technology (20191017).

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
