# Peer review of "Snow cover variations across China from 1952-2012"

_The Cryosphere, 2019_

## Referee Comment (RC1) · Anonymous Referee #1 · 22 Oct 2019

This manuscript characterizes snow cover variability and trends across three regions of China through the analysis of the surface snow depth observation network. While the subject is relevant to The Cryosphere and represents a potentially important contribution to the assessment of regional snow cover trends, there are some critical shortcomings.

1. The end date of 31 December 2013 is problematic. It makes more sense to use the end of a snow season or hydrological year. Small point, but because the complete 2012-2013 season was available for analysis, should the title not use the year range 1952-2013? More importantly, why was the seemingly arbitrary decision made to only utilize data through 2013, which means the reported trends do not include the six most recent winters. This means the reported trends have reduced relevancy and timeliness.

[Figure]

As noted below, there were some extreme snow events in China during the 2009-2012 time period, so it would be interesting to see if these strong anomalies continue to emerge between 2013 and 2019.

2. Lines 85-92: It's not clear how time series of varying length at individual measurement locations were utilized in the analysis. Do the map figures (e.g. Figures 2, 4, and 6) show all stations with at least 10 years of data as noted on line 86? If so, does this mean there are different start and end dates for individual stations in these figures, or do all points over the same time period? Do some points represent trends calculated over only 10 years of data? In the figures which show regional averages (e.g. Figures 3, 5, and 7) how was variability in station time series length accounted for in the averaging procedure? More details on how the snow data were handled is needed in the first part of Section 2.1.

3. Some of the analyzed snow variables are consistent with typical metrics of assessing snow cover, such as maximum snow depth, snow cover days, snow onset, snow end days, etc. but some of the snow variables are unusual and lack a clear physical driver that can be related to climate variability. I suggest dropping annual average snow depth and cumulative snow depth. In the case of snow onset date (SOD) and snow end date (SED), do these mark the beginning and end of continuous snow cover, or do they include ephemeral snow cover events (such as the first time snow accumulates, even if it melts again before continuous snow cover begins)?

4. Section 2: the text describing Mann-Kendall and linear trends in parts (a) and (b) is not easily readable – it's essentially a series of equations which describe fairly straightforward statistical techniques. Sections (c) and (d) however provide only very short descriptions of more complicated techniques. Should the wavelet analysis and structural equation modeling be retained in the paper (see comment #8 below) the text in parts (c) and (d) must be expanded, while the readability of parts (a) and (b) should be improved.

5. Section 3.1: I think most readers will not be clear on the terms UF and UB as utilized in the text. The definition of UF can be discerned from Section 2.1, but this is not the case for UB.

6. Figures 3 and 5: the strength of these trends seem to be driven strongly by the early and late years in the time series. Without a handful of large anomalies during the first five years and the last five years, the trends appear to be near zero. Can some comments be added on the influence of these outliers at the beginning and end of the time series?

7. I'm confused by the determination of some of the break points in the time series. For example, in Table 2 – how can some regions and variables have break points in consecutive years (such as 1956 and 1957)?

8. Many of the overall trends are quite modest (e.g a decrease in snow cover days of 0.1 days per decade, which translate to not even a full day over the 70-year period of record). The lack of trends is itself not an issue – it's still useful to report this in the literature given the long time period. But with the lack of strong trends evident in the Mann-Kendall analysis, I don't believe the wavelet analysis and structural equation modeling are warranted. My suggestion would be to remove this analysis – Figures 8 and 9 are difficult to interpret and don't add much insight. Instead, some additional analysis could add more value and innovation to the paper. I was left wondering two things: -Are these trends consistent with satellite-derived data such as the NOAA snow chart climate data record (https://climate.rutgers.edu/snowcover/index.php) which extend back to 1977? -Can snow cover trends over China be attributed to temperature and/or precipitation trends? In general, it appears the snow cover season is getting shorter (Figure 7) but the snow depth is increasing (Figure 3). How do the spatial patterns of seasonal temperature changes and precipitation change (including the proportion of snow versus rain) compare to the station observations of snow?

A couple minor comments: -Line 17: the term 'jump points' is used here and throughout

the manuscript, but I think 'break points' is a more appropriate term -Figure 1: Given the emphasis placed on elevation for some of the interpretation, can shading be added to this figure to show elevation?

The manuscript also requires a thorough edit for grammar, English usage, and word choice. Edits of this nature were too numerous to identify individually in this review.

---

## Referee Comment (RC2) · Ronan Connolly (Referee) · 30 Oct 2019

**General comments** The authors have used the Chinese Meteorological Administration (CMA)'s daily snow depth dataset for China that covers the period, 1951-2013, to study and describe the annual variations in snow cover across China over the period, 1952-2012 (because they define the hydrological year as July-June, they were unable to include 1951 or 2013). This dataset is constructed from ground-based meteorological measurements taken from hundreds of meteorological stations distributed across most of China. The authors derive several different metrics of snow cover from this dataset and compare and contrast the annual trends for each of these metrics. They also describe separately the trends for the three main snow cover areas in China (northeast China; northern Xinjiang; and the Tibetan Plateau) as well as the overall trends for all

[Figure]

China.

Although it is disappointing that the dataset ends in 2013 (and the analysis ends in 2012), the results should still be of interest to readers of The Cryosphere, and more broadly, the wider climate science community. This is especially so for three reasons:

1. Much of the analysis of snow cover trends described in the literature is based on satellite-derived estimates (in particular, the "Rutgers Snow Lab" dataset, which covers the period 1967-present), while the analysis in this paper is based on an independent ground-based dataset.

2. The snow cover trends for China have received some discussion and debate in the literature lately because they appear to be different from much of the Northern Hemisphere, e.g., see Wei and Dong, 2015; Wang et al., 2017; Chen et al., 2016.

3. Recently, some of us (and others) have been encouraging more research into the differences in climatic trends in different regions within China, e.g., Soon et al., 2018; Li and Yang, 2019; Soon et al., 2019. Therefore, this study is also useful because the authors provide the breakdown for the trends of each of the three main snow cover regions in China as well as the national trends.

However, in my opinion, the authors should compare and contrast their findings with other equivalent datasets. At a minimum, the authors should compare their results to:

1. The satellite-derived snow cover extent dataset maintained by Rutgers University Global Snow Lab (Robinson et al., 1993; Robinson and Frei, 2000; Estilow et al., 2015): https://climate.rutgers.edu/snowcover/docs.php?target=datareq

   Ideally, if the authors are used to working with NetCDF files, they could extract the gridded trends for China (and perhaps even the three regions) from the gridded dataset. However, if not, they should at least compare their results to the Northern

Hemisphere trends over the common period, i.e., 1967-2012. They might also want to comment on the post-2012 trends that are available from the Rutgers dataset since their own analysis finishes in 2012.

2. How do their results compare with the CMIP5 and/or CMIP6 Global Climate Model (GCM) hindcasts for China? KNMI have a useful website which provides access to many of the CMIP5 hindcast results (including "snow cover area"), and if you register on the website, you can apply a "country land mask" to the results, allowing you to extract the hindcasted regional trends for China. https://climexp.knmi.nl/start.cgi

   The Climate Explorer website sometimes has intermittent access, since it is largely the part-time efforts of just one researcher, Prof. van Oldenborgh. However, it is a very versatile website, and if the authors don't have access to CMIP5 or CMIP6 data from elsewhere, they could probably use this to extract the results for the CMIP5 hindcasts. For instance, we recently used this website for our analysis of Northern Hemisphere snow cover trends in Connolly et al., 2019.

They should also discuss in more detail how their results compare to other studies analysing Chinese snow cover (e.g., the Wei and Dong, 2015; Wang et al., 2017; Chen et al., 2016 papers mentioned above), as well as to the trends for the rest of the Northern Hemisphere (e.g., see the Connolly et al., 2019 paper mentioned above).

In a couple of places, the authors hint at how their analysis could be of relevance for anthropogenic global warming, and they uncritically paraphrase a claim from the IPCC's 5th Assessment Report. However, they don't seem to provide any actual discussion of the relevance. Indeed, as I will discuss below, if anything, their results are problematic for the IPCC's claims on snow cover. For this reason, the authors should *either* drop the discussion of anthropogenic global warming *or else* provide a more critical evaluation of how their results compare/contrast with the IPCC report (and other literature, e.g., Connolly et al., 2019).

With that in mind, if the authors were to satisfactorily do all of the above, then the manuscript would be worthy of publication in The Cryosphere. I will provide some additional comments that are more detailed below.

**Specific comments**

**1.)** *Relevance for anthropogenic global warming?*

On lines 50-53, you state, "The Intergovernmental Panel on Climate Change (IPCC) has reported that climate warming over the past 50 years is indisputable and that the temperature over the past 50 years is likely to be the highest on average over the past 500 years (IPCC, 2013)". There are several problems here:

- a) There is actually no "(IPCC, 2013)" listed in the references. Instead, the IPCC AR5 Working Group 1 report is currently listed as Stocker et al., 2013.

- b) It is unclear exactly which specific part of the >1000 page IPCC reports the authors are referring to. There are many different *similar-sounding* claims made in the report, but the IPCC actually often make *very* specific and precise claims that have been carefully parsed to simplify their narrative.

  Often if the scope of a particular IPCC claim were broadened to describe the bigger picture, it would make their narrative less compelling. For this reason, the IPCC lead authors are usually very precise in what specifically they are claiming.

  For instance, on p7 of the Summary for Policymakers, the IPCC AR5 WG1 claims, "Over the last two decades, *[...]* **Arctic** sea ice and Northern Hemisphere **spring** snow cover have continued to decrease in extent (high confidence) (see Figure SPM.3). 4.2–4.7" (emphasis added in bold).

  Antarctic sea ice trends over the same period had actually increased, as had Northern Hemisphere winter and autumn snow cover. However, by not mentioning these results, the IPCC were able to create a more compelling narrative of an

overall "warming of the climate system" (p2, Summary for Policymakers), without making any false claims. In other words, their claims (especially the ones made in the Summary for Policymakers) are **not** made to accurately inform the scientific community of the full context of their findings, but rather to selectively present information which makes their overall narrative seem as compelling as possible.

As an aside, you might have guessed from the above that I'm **not** particularly impressed by the scientific rigour of the IPCC reports. This is true. However, regardless of what you think of the IPCC reports, my point is that, when quoting the IPCC reports, you have to be *very careful* in their specific quotations. Otherwise their statements could be inaccurate.

With that in mind, in the Summary for Policymakers, the IPCC WG1 AR5 claims, "Each of the last three decades has been successively warmer at the Earth's surface than any preceding decade since 1850 (see Figure SPM.1). In the Northern Hemisphere, 1983–2012 was likely the warmest 30-year period of the last 1400 years (medium confidence)." (Summary for Policymakers, p3) Is that the specific claim you are referring to, or is it something else? Because it is not quite the same as the statement you made.

- c) With regards to the claim which you attribute to the IPCC, it is debatable, and could seem somewhat cherry-picked. Most paleoclimate temperature reconstructions argue that the 17th, 18th and 19th centuries were relatively cold ("Little Ice Age"), but that roughly 1,000 years ago, there was a relatively warm period ("Medieval Warm Period").

There is, of course, considerable debate over how the Current Warm Period compares to the Medieval Warm Period. The IPCC argues that the Current Warm Period is "likely" warmer. But, notice that even in the IPCC's claim that I quoted above, they only assign medium confidence and the term "likely" (66-100% chance according to the IPCC "likelihood scale") to their claim that "1983-2012 was likely the warmest 30-year period of the last 1400 years" (and that they

confine this statement to the Northern Hemisphere).

In other words, the IPCC have actually left it as fairly plausible (up to 34% chance according to the IPCC) that it was similarly warm during the Medieval Warm Period. So, choosing "the past 500 years" as a benchmark (i.e., excluding the Medieval Warm Period) might seem like cherry picking.

Even if you had stuck to the shorter instrumental period (i.e., the last 150 years or so), there is debate over the relative warmth of the recent warm period to the early 20th century warm period. In Soon et al. (2015), we argue that the current land surface temperature datasets have failed to satisfactorily correct for non-climatic biases, including urbanization bias. We developed an alternative estimate of Northern Hemisphere temperature trends since 1881 using mostly rural stations. This new estimate suggests that, while temperatures increased from the 1970s to 2000s, they cooled from the 1940s to 1970s, and temperatures in the 1940s were comparable to present. This could contradict your claim (which you attributed to the IPCC) that the most recent 50 years were the hottest over the past 500 years.

More recently, in Soon et al. (2018), we reviewed the more relevant debate over how the current warm period compares to the early 20th century warm period **for China**. We showed that there is considerable debate over how warm the earlier warm period is relative to the current warm period. Indeed, this debate is ongoing, e.g., see Li and Yang (2019) and our reply (Soon et al., 2019). However, hopefully this should illustrate how you need to be very careful in making these sort of generic statements, when you are referring to the IPCC!

- d) Moreover, it is unclear, why you are making this claim anyway. Later (lines 69-70), you add, "In the context of global warming, the feedback of the snow cover in China on climate change is unknown.".

I appreciate that it has become "fashionable" to include a reference to "global
warming" or "global climate change" in *any* paper looking at climate trends. How-ever, if you really want to frame your results in the context of global warming (as opposed to framing your paper as a study of regional climate change), then you should be more rigorous, and show the **full** context.

For instance, the IPCC claim on p2 of the Summary for Policymakers that, "Warm-ing of the climate system is unequivocal, and since the 1950s, many of the ob-served changes are unprecedented over decades to millennia. The atmosphere and ocean have warmed, **the amounts of snow and ice have diminished**, sea level has risen, and the concentrations of greenhouse gases have increased (see Figures SPM.1, SPM.2, SPM.3 and SPM.4). 2.2, 2.4, 3.2, 3.7, 4.2–4.7, 5.2, 5.3, 5.5–5.6, 6.2, 13.2" (emphasis added in bold). Then, on p7, they claim that, "Over the last two decades, *[...]* Northern Hemisphere spring snow cover *[has]* contin-ued to decrease in extent (high confidence) (see Figure SPM.3). 4.2–4.7".

Yet, this paper shows that, **for China**, the amounts of (annual) snow cover have generally increased.

You are not the first to notice this difference for China (e.g., Wei and Dong, 2015; Wang et al., 2017; Chen et al., 2016 as mentioned above). And, in Connolly et al. (2019), we showed that the IPCC's "spring snow cover" claim was not representative of the overall snow cover trends, e.g., winter and autumn snow cover seems to have generally increased over the same period.

With that in mind, I recommend you go with one of two options:

- Option 1: Explain how your results are somewhat at odds with the IPCC AR5's claims and discuss the full context.

  *or*

- Option 2: Drop the references to "global warming" and stick to describing the trends for China.
**2.)** *Questions on the datasets*

- a) Do you know if the CMA are planning on updating their snow depth dataset to include more recent years? If not, could you include a comment on why it has not been updated since 2013?

- b) On lines 84-87, you say, "To ensure the reasonableness of statistical analysis, we must ensure that the station dataset used for statistical analysis were longer than 10 years. Therefore, the stations with less than 10 years of records were omitted from the analysis.". Can you elaborate on how many stations you had in total and how many were omitted after these steps?

- c) Will your results be available as Supplementary Information? If so, this would be great.

  Ideally, as well as the time series you constructed, I would also like to see the individual station results available as Supplementary Information. But, if that is not possible (since it is a CMA dataset), it would be helpful if you could at least provide some indication of the numbers (and possibly locations) of the stations available for each year.

- d) On line 79, you say that the dataset begins on 1 January 1951, but then on line 90, you say, "...since the SD measurements began in October 1951". Which date is it?

**3.)** *Results*

- a) Should the title of Section 3 be "Results" instead of "Result"?

- b) On lines 206-207, you say, "The national $SD_{average}$, $SD_{overall}$ and $SD_{max}$ showed increasing trends from 1952 to 2012 under the context of global warming". What do you mean by this?

First, as explained above, I would reconsider whether or not you want to frame this paper in the context of global warming.

Also, do you mean "anthropogenic global warming", i.e., an expected global warming trend arising from increasing greenhouse gas concentrations? If so, then as we discussed in Connolly et al. (2019), the CMIP5 hindcasts (which do indeed attribute most of the global temperature trends since the 1950s to anthropogenic greenhouse gas emissions) predict that Northern Hemisphere snow cover should have been **decreasing**, and **not** increasing as you find. In other words, your results are actually somewhat problematic "in the context of (anthropogenic) global warming".

- c) But, instead of focusing on global trends, why not frame these results simply in the context of *regional* climate trends for China? How do your results compare with the equivalent temperature and precipitation trends for China?

In this context, we provided a review of Chinese temperature trends in Soon et al. (2018). You might also find the ensuing discussion, i.e., Li and Yang (2019) and Soon et al. (2019) relevant.

- d) As mentioned in the General Comments section, you should include at least some comparison with the Rutgers satellite-derived dataset (for example). Ideally, I would calculate the regional trends for China from their gridded dataset. But, at the very least, your results should be compared to the overall Northern Hemisphere time series.

- e) As also mentioned in the General Comments section, you should compare how your results compare to the CMIP5 and/or CMIP6 hindcasts of snow cover for China. See Connolly et al. (2019) for a systematic comparison of Northern Hemisphere snow cover according to observations vs. all available CMIP5 model runs.

- f) Have you looked at the differences between seasons? As we discussed in Connolly et al. (2019), the trends for Northern Hemisphere are quite different for each season. Depending on how you carried out your analysis, this might be beyond the scope of your paper, but you should suggest it as a possibility for future work if you think that this could be done using your dataset.

- g) To be honest, most of the statistical analysis in Sections 3.4 and 3.5 seems unnecessary to me and doesn't really add much insight (in my opinion). Personally, I would remove these two sections. However, perhaps some readers might find it of interest, so I will leave it up to you whether you think these sections are needed or not.

**4.)** *Conclusion*

Your Section 5 currently reads more like a "summary" than a conclusions. It is ok to include a brief summary as part of the conclusions, but you should **also** provide some concluding remarks and/or some recommendations for further research.

**Technical corrections**

The written English is not great with some grammatical errors scattered throughout the manuscript. I suggest using the "Check spelling and grammar" option of Microsoft Word or some other similar word processor, and fixing any of the underlined errors that will appear.

Review by Dr. Ronan Connolly

**References cited in this review**

*Note: I am a co-author of several of the papers listed below. However, I dislike when a reviewer uses their review as an excuse to promote their own papers. For this reason, the authors are under no obligation to cite any of my listed papers, but I'm including them in case the authors do find them relevant!*

- Chen, Liang, Cao and He, 2016. Distribution, attribution, and radiative forcing of snow cover changes over China from 1982 to 2013. Climatic Change, 137, 363-377, doi: 10.1007/s10584-016-1688-z

- Connolly, Connolly, Soon et al., 2019. Northern Hemisphere snow-cover trends (1967-2018): a comparison between climate models and observations. Geosciences, vol. 9, 135, doi: 10.3390/geosciences9030135

- Estilow, Young and Robinson, 2015. A long-term Northern Hemisphere snow cover extent data record for climate studies and monitoring. Earth System Science Data, 7, 137-142, doi:10.5194/essd-7-137-2015.

- Li and Yang, 2019. Comments on "Comparing the current and early 20th century warm periods in China" by Soon W., R. Connolly, M. Connolly et al. Earth-Science Reviews, in press, doi: 10.1016/j.earscirev.2019.102886

- Robinson and Frei, 2000. Seasonal variability of northern hemisphere snow extent using visible satellite data. Professional Geographer, 51, 307-314, doi:10.1111/0033-0124.00226.

- Robinson, Dewey and Heim, Jr., 1993. Global snow cover monitoring: an update. Bulletin of the American Meteorological Society, 74, 1689-1696, doi:10.1175/1520-0477(1993)074<1689:GSCMAU>2.0.CO;2

- Soon, Connolly and Connolly, 2015. Re-evaluating the role of solar variability on Northern Hemisphere temperature trends since the 19th century. Earth-Science Reviews, vol. 150, 409-452, doi: 10.1016/j.earscirev.2015.08.010

- Soon, Connolly, Connolly et al., 2018. Comparing the current and early 20th century warm periods in China. Earth-Science Reviews, vol. 185, 80-101, doi: 10.1016/j.earscirev.2018.05.013

[Figure]

- Soon, Connolly, Connolly et al., 2019. Reply to Li & Yang's comments on "Comparing the current and early 20th century warm periods in China". Earth-Science Reviews, in press, doi: 10.1016/j.earscirev.2019.102950

- Wei and Dong, 2015. Assessment of simulations of snow depth in the Qinghai-Tibetan Plateau using CMIP5 multi-models. Arctic, Antarctic, and Alpine Research, 47, 611-625, doi: 10.1657/AAAR0014-050

- Wang, Wu, Wang et al., 2017. No evidence of widespread decline of snow cover on the Tibetan Plateau over 2000-2015, Scientific Reports, 7, 14645, doi: 10.1038/s41598-017-15208-9

---

## Editor Comment (EC1) · Florent Dominé (Editor) · 31 Oct 2019

Dear Authors,

Your paper definitely provides interesting data relevant to topics relevant to the Cryosphere. However the reviewers recommends major changes that you will need to address in detail before possible publication. Please pay special attention regarding the time span studied. Reviewer 1 wonders why your analysis does not use data until 2019? This is important for a paper submitted in 2019, and also to evaluate the significance of the unusual data in the last years of your study. Reviewer 2 also raises this point and a number of other issues, in particular your discussion of IPCC conclusions. Please explain how you plan to address these comments before submitting a revised

version. Finally, it is usual practice for papers published in TC to make their dataset readily available. You do mention that your data were obtained from the extensive Chinese Meteorological Data Service Center (http://data.cma.cn/en) but a readily useable data set would certainly be appreciated by the community. Reviewer 2 also raises this issue.

Thank you for submitting your interesting paper to The Cryosphere. I look forward to reading your response to the reviewers' constructive comments.

Best regards

Florent Domine

---

## Author Comment (AC1) · 28 Nov 2019

**1.** **Comments from Referees:** The end date of 31 December 2013 is problematic. It makes more sense to use the end of a snow season or hydrological year. Small point, but because the complete 2012-2013 season was available for analysis, should the title not use the year range 1952-2013? More importantly, why was the seemingly arbitrary decision made to only utilize data through 2013, which means the reported trends do not include the six most recent winters. This means the reported trends have reduced relevancy and timeliness. As noted below, there were some extreme snow events in China during the 2009-2012 time period, so it would be interesting to see if these strong anomalies continue to emerge between 2013 and 2019.

**Author's Response:** Firstly, on behalf of all authors, we appreciate your affirmation and also great comments for this manuscript. Based on your suggestion, the snow data from 2014 to 2019 is collected through our effects, which extend the period from 1951 to 2018. The hydrological year which spanned from July 1 of the current year to June 31 of the subsequent year is used when analyze the snow variations in this study, thus, the data period used between July 1, 1951 and June 30, 2018. The title of the manuscript is modified as: "Snow cover variations across China from 1951-2018"

**Author's changes in manuscript:** The data and results are re-analyzed based on extended dataset.

**2.** **Comments from Referees:** Lines 85-92: It's not clear how time series of varying length at individual measurement locations were utilized in the analysis. Do the map figures (e.g. Figures 2, 4, and 6) show all stations with at least 10 years of data as noted on line 86? If so, does this mean there are different start and end dates for individual stations in these figures, or do all points over the same time period? Do some points represent trends calculated over only 10 years of data? In the figures which show regional averages (e.g. Figures 3, 5, and 7) how was variability in station time series length accounted for in the averaging procedure? More details on how the snow data were handled is needed in the first part of Section 2.1.

**Author's Response:** The proportion of length records in years for each stations during the period of 1951 to 2018 are calculated and mapped in Fig 1 based on your suggestion. There 67

stations which the snow records less than 10 years were abandoned in this study based on data quality control. More details on how the snow the snow data handled is described in section 2.1. Thank you for your comments.

**Author's changes in manuscript:** Please see the section 2.1 about data description and the figure 1.

[Figure]

**Figure 1.** Geographical locations and the proportion of length records in years of each meteorological stations in China mainland.

**3.    Comments from Referees:** Some of the analyzed snow variables are consistent with typical metrics of assessing snow cover, such as maximum snow depth, snow cover days, snow onset, snow end days, etc. but some of the snow variables are unusual and lack a clear physical driver that can be related to climate variability. I suggest dropping annual average snow depth and cumulative snow depth. In the case of snow onset date (SOD) and snow end date (SED), do these mark the beginning and end of continuous snow cover, or do they include ephemeral snow cover events (such as the first time snow accumulates, even if it melts again before continuous snow cover begins)?

**Author's Response:** We are agree with you. The accumulative snow depth lack a clear physical drive that can be related to climate, and also hard to explore its variation. But the annual mean snow depth ($SD_{overall}$) can somehow reflect the interannual variation of snow accumulation.

Thus, the accumulative snow depth is dropped based on your suggestion. In order to avoid the impact of ephemeral snow in snow phenology computations, the SOD is defined as the first date of the first three continuous snow records during the snow accumulation period, and the SED is defined as the last day of the last three continuous snow records during the snow melt season, respectively. Thank you for the comment.

**Author's changes in manuscript:** Please see the revised manuscript in page 103-115.

**4.    Comments from Referees:** Section 2: the text describing Mann-Kendall and linear trends in parts (a) and (b) is not easily readable – it's essentially a series of equations which describe fairly straightforward statistical techniques. Sections (c) and (d) however provide only very short descriptions of more complicated techniques. Should the wavelet analysis and structural equation modeling be retained in the paper (see comment #8 below) the text in parts (c) and (d) must be expanded, while the readability of parts (a) and (b) should be improved.

**Author's Response:** Thank you for the comment. We are sorry that we didn't give more information for our methodologies. In the revised manuscript, the detail description for UF and UB in M-K test, as well as wavelet. Thank you again.

**Author's changes in manuscript:** Please see the section 2.2.

**5.    Comments from Referees:** Section 3.1: I think most readers will not be clear on the terms UF and UB as utilized in the text. The definition of UF can be discerned from Section 2.1, but this is not the case for UB.

**Author's Response:** More detail definition for UF and UB are described in Section 2.2. Thank you for the comments.

**Author's changes in manuscript:** Please see the section 2.2.

**6.    Comments from Referees:** Figures 3 and 5: the strength of these trends seem to be driven strongly by the early and late years in the time series. Without a handful of large anomalies during the first five years and the last five years, the trends appear to be near zero. Can some comments be added on the influence of these outliers at the beginning and end of the time series?

**Author's Response:** This anomalies may be caused by factors such as temperature, participation and the oscillation period of the snow cover. But we don't have the supporting data to prove it. Sorry for the response, and thank you for the comment.

**Author's changes in manuscript:** Please see the section 3.4.

**8.    Comments from Referees:** Many of the overall trends are quite modest (e.g a decrease in snow cover days of 0.1 days per decade, which translate to not even a full day over the 70-year period of record). The lack of trends is itself not an issue – it's still useful to report this in the literature given the long time period. But with the lack of strong trends evident in the Mann-Kendall analysis, I don't believe the wavelet analysis and structural equation modeling are warranted. My suggestion would be to remove this analysis – Figures 8 and 9 are difficult to interpret and don't add much insight. instead, some additional analysis could add more value and innovation to the paper. I was left wondering two things: -Are these trends consistent with satellite-derived data such as the NOAA snow chart climate data record (https://climate.rutgers.edu/snowcover/index.php) which extend back to 1977? -Can snow cover trends over China be attributed to temperature and/or precipitation trends? In general, it appears the snow cover season is getting shorter (Figure 7) but the snow depth is increasing (Figure 3). How do the spatial patterns of seasonal temperature changes and precipitation change (including the proportion of snow versus rain) compare to the station observations of snow?

**Author's Response:** We are completely agreed with your comments. Based on your comments, The wavelet analysis and structural equation modeling results are deleted. The snow variations trends are compared with the results from remote sensing observations as well as the simulation results from CMIP5 model. We rewrote the discussion section, and focus on the consistent with other current results to further prove the findings in this study. Unfortunately, we don't have climate data to prove the snow cover trends which conclusions drawn from this article. In any case, we still appreciate your great suggestion.

**Author's changes in manuscript:** Please see the detail discussion in section 4.

**9.    Comments from Referees:** A couple minor comments: -Line 17: the term 'jump points' is used here and throughout the manuscript, but I think 'break points' is a more appropriate term -Figure 1: Given the emphasis placed on elevation for some of the interpretation, can shading be added to this figure to show elevation?

**Author's Response:** The 'jump points' are revised to 'break points' based on your suggestion. For figure 1, because of the too many colors used to distinguish snow zones and meteorological stations, the use of shading relief as the background will affect the reading of the figure. We

tried many schemes, but can't find an ideal way. Thus the original scheme was kept. We are sorry we can't meet your comment. Please forgive us.

**New references updated:**

Brown, R. D., and Robinson, D. A.: Northern Hemisphere spring snow cover variability and change over 1922–2010 including an assessment of uncertainty. Cryosphere, 5, 219–229, doi:10.5194/tc-5-219-2011, 2011

Chen, X., Liang, S., Cao, Y., and He, T.: Distribution, attribution, and radiative forcing of snow cover changes over China from 1982 to 2013. Clim.Change.137, 363–77, doi:10.1007/s10584-016-1688-z, 2016.

Choi, G., Robinson, D., and Kang, S.: Changing Northern Hemisphere snow seasons. J.Clim. 23, 5305–5310, doi:10.1175/2010JCLI3644.1, 2010.

Connolly, R., Connolly, M., Soon, W., Legates, D. R., Cionco, R. G., Velasco Herrera, V. M.: Northern Hemisphere snow-cover trends (1967-2018): A comparison between climate models and observations. Geosci., 9, 135, doi:10.3390/geosciences9030135, 2019.

Derksen, C., and Brown, R.: Spring snow cover extent reductions in the 2008-2012 period exceeding climate model projections. Geophys. Res. Lett., 39, L19504, doi: 10.1029/2012GL053387, 2012.

Euskirchen, E. S., McGuire, A. D., Chapin, F.S.: Energy feedbacks of northern high-latitude ecosystems to the climate system due to reduced snow cover during 20th century warming. Glob. Chang. Biol. 13, 2425–2438, doi:10.1111/j.1365-2486.2007.01450.x, 2007.

Hall, D.K., Riggs, G.A., Salomonson, V.V., DiGirolamo, N.E., Bayr, K.J. MODIS snow-cover products. Remote Sens. Environ. 83, 181–194, doi: 10.1016/S0034-4257(02)00095-0, 2002.

Hori, M., Sugiura, K., Kobayashi, K., Aoki, T., Tanikawa, T., Kuchiki, K., Niwano, M., Enomoto, H.: A 38-year (1978–2015) Northern Hemisphere daily snow cover extent product derived using consistent objective criteria from satellite-borne optical sensors, Remote. Sens. Environ., 191, 402-418, 2017.

Peng, S., Piao, S., Ciais, P., Friedlingstein, P., Zhou, L., Wang, T.: Change in snow phenology and its potential feedback to temperature in the Northern Hemisphere over the last three decades, Environ. Res. Letters., 8, 014008, 2013.

Percival, D. B., Walden, A. T.: Wavelet Methods for Time Series Analysis (Cambridge Series in Statistical and Probabilistic Mathematics), UK: Cambridge Univ. Press, 2000.

Wei, Z., and Dong, W.: Assessment of Simulations of Snow Depth in the Qinghai-Tibetan Plateau Using CMIP5 Multi-Models, Arctic Antarct. & Alp. Res., 47, 611-525, 2015.

---

## Author Comment (AC2) · 28 Nov 2019

**General comments**

The authors have used the Chinese Meteorological Administration (CMA)'s daily snow depth dataset for China that covers the period, 1951-2013, to study and describe the annual variations in snow cover across China over the period, 1952-2012 (because they define the hydrological year as July-June, they were unable to include 1951 or 2013). This dataset is constructed from ground-based meteorological measurements taken from hundreds of meteorological stations distributed across most of China. The authors derive several different metrics of snow cover from this dataset and compare and contrast the annual trends for each of these metrics. They also describe separately the trends for the three main snow cover areas in China (northeast China; northern Xinjiang; and the Tibetan Plateau) as well as the overall trends for all China. Although it is disappointing that the dataset ends in 2013 (and the analysis ends in 2012), the results should still be of interest to readers of The Cryosphere, and more broadly, the wider climate science community. This is especially so for three reasons:

1. Much of the analysis of snow cover trends described in the literature is based on satellite-derived estimates (in particular, the "Rutgers Snow Lab" dataset, which covers the period 1967-present), while the analysis in this paper is based on an independent ground-based dataset.

2. The snow cover trends for China have received some discussion and debate in the literature lately because they appear to be different from much of the Northern Hemisphere, e.g., see Wei and Dong, 2015; Wang et al., 2017; Chen et al., 2016.

3. Recently, some of us (and others) have been encouraging more research into the differences in climatic trends in different regions within China, e.g., Soon et al., 2018; Li and Yang, 2019; Soon et al., 2019. Therefore, this study is also useful because the authors provide the breakdown for the trends of each of the three main snow cover regions in China as well as the national trends.

However, in my opinion, the authors should compare and contrast their findings with other equivalent datasets. At a minimum, the authors should compare their results to:

1. The satellite-derived snow cover extent dataset maintained by Rutgers University Global Snow Lab (Robinson et al., 1993; Robinson and Frei, 2000; Estilow et al., 2015): https://climate.rutgers.edu/snowcover/docs.php?target=datareq. Ideally, if the authors are used to working with NetCDF files, they could extract the gridded trends for China (and perhaps even the three regions) from the gridded dataset. However, if not, they should at least compare their results to the Northern Hemisphere trends over the common period, i.e., 1967-2012. They might also want to comment on the post-2012 trends that are available from the Rutgers dataset since their own analysis finishes in 2012.

2. How do their results compare with the CMIP5 and/or CMIP6 Global Climate Model (GCM) hindcasts for China? KNMI have a useful website which provides access to many of the CMIP5 hindcast results (including "snow cover area"), and if you register on the website, you can apply a "country land mask" to the results, allowing you to extract the hindcasted regional trends for China. https://climexp.knmi.nl/start.cgi.

The Climate Explorer website sometimes has intermittent access, since it is largely the part-time efforts of just one researcher, Prof. van Oldenborgh. However, it is a very versatile website, and if the authors don't have access to CMIP5 or CMIP6 data from elsewhere, they could probably use this to extract the results for the CMIP5 hindcasts. For instance, we recently used this website for our analysis of Northern Hemisphere snow cover trends in Connolly et al., 2019.

They should also discuss in more detail how their results compare to other studies analysing Chinese snow cover (e.g., the Wei and Dong, 2015; Wang et al., 2017; Chen et al., 2016 papers mentioned above), as well as to the trends for the rest of the Northern Hemisphere (e.g., see the Connolly et al., 2019 paper mentioned above).

In a couple of places, the authors hint at how their analysis could be of relevance for anthropogenic global warming, and they uncritically paraphrase a claim from the IPCC's 5th Assessment Report. However, they don't seem to provide any actual discussion of the relevance. Indeed, as I will discuss below, if anything, their results are problematic for the IPCC's claims on snow cover. For this reason, the authors should either drop the discussion of anthropogenic global warming or else provide a more critical evaluation of how their results compare/contrast with the IPCC report (and other literature, e.g., Connolly et al., 2019).

With that in mind, if the authors were to satisfactorily do all of the above, then the manuscript would be worthy of publication in The Cryosphere. I will provide some additional comments that are more detailed below.

**Author's Response:**

Dear Prof. Connolly,

On behalf of all author of this manuscript, we appreciate your great comments and also you careful review. From your comments, we know you are so professional scientist in research on long-term meteorological elements with very cautious attitude. Please accept our respect and gratitude to you for your pertinent suggestion and responsible review.

Base on your comments, the revised manuscript has made the following changes:

1) The snow data from 2014 to 2019 is collected through our effects, which extend the period from 1951 to 2018. We recalculated and rewrote the results and discussion sections in the revised manuscript.

2) Cause we don't have the supporting data like temperature and precipitation to prove our results in this manuscript. So the reference involved in "global warming" were dropped, and stick to describe the snow cover trends for China only. Furthermore, the trends pattern in China so different with Northern Hemisphere, the reasons should be further reveal combine the meteorological data in future. Our previous published paper also proved the differences:

Huang X., Deng, L., Ma, X., Wang Y., Hao, X., and Liang, T.: Spatiotemporal dynamics of snow cover based on multi-source remote sensing data in China, Cryosphere, 10, 2453-2463, doi:10.5194/tc-10-2453-2016, 2016.

Wang, Y., Huang, X., Liang, H., Sun, Y., Feng, Q., and Liang, T.: Tracking Snow Variations in the Northern Hemisphere Using Multi-Source Remote Sensing Data (2000–2015), Remote Sensing, 10, 136, doi:10.3390/rs10010136, 2018.

3) The snow variations trends are compared with the results from remote sensing observations as well as the simulation results from CMIP5 model. We definitely know the depth in comparison is far from your article in Connolly et al. 2019. Cause our less understanding in numerical simulation like CMIP5. We rewrote the discussion section, and focus on the consistent with other current results to further prove the findings in this study.

Other response for your comments are follows:

**1. Comments from Referees:**

Relevance for anthropogenic global warming?

On lines 50-53, you state, "The Intergovernmental Panel on Climate Change (IPCC) has reported that climate warming over the past 50 years is indisputable and that the temperature over the past 50 years is likely to be the highest on average over the past 500 years (IPCC, 2013)". There are several problems here:

a) There is actually no "(IPCC, 2013)" listed in the references. Instead, the IPCC AR5 Working Group 1 report is currently listed as Stocker et al., 2013.

b) It is unclear exactly which specific part of the >1000 page IPCC reports the authors are referring to. There are many different similar-sounding claims made in the report, but the IPCC actually often make very specific and precise claims that have been carefully parsed to simplify their narrative.

Often if the scope of a particular IPCC claim were broadened to describe the bigger picture, it would make their narrative less compelling. For this reason, the IPCC lead authors are usually very precise in what specifically they are claiming.

For instance, on p7 of the Summary for Policymakers, the IPCC AR5 WG1 claims, "Over the last two decades, [...] **Arctic sea ice** and **Northern Hemisphere spring snow cover** have continued to decrease in extent (high confidence) (see Figure SPM.3). 4.2–4.7" (emphasis added in bold).

Antarctic sea ice trends over the same period had actually increased, as had Northern Hemisphere winter and autumn snow cover. However, by not mentioning these results, the IPCC were able to create a more compelling narrative of an overall "warming of the climate system" (p2, Summary for Policymakers), without making any false claims. In other words, their claims (especially the ones made in the Summary for Policymakers) are not made to accurately inform the scientific community of the full context of their findings, but rather to selectively present information which makes their overall narrative seem as compelling as possible.

As an aside, you might have guessed from the above that I'm not particularly impressed by the scientific rigour of the IPCC reports. This is true. However, regardless of what you think of the IPCC reports, my point is that, when quoting the IPCC reports, you have to be very careful in their specific quotations. Otherwise their statements could be inaccurate.

With that in mind, in the Summary for Policymakers, the IPCC WG1 AR5 claims, "Each of the

last three decades has been successively warmer at the Earth's surface than any preceding decade since 1850 (see Figure SPM.1). In the Northern Hemisphere, 1983–2012 was likely the warmest 30-year period of the last 1400 years (medium confidence)." (Summary for Policymakers, p3) Is that the specific claim you are referring to, or is it something else? Because it is not quite the same as the statement you made.

c) With regards to the claim which you attribute to the IPCC, it is debatable, and could seem somewhat cherry-picked. Most paleoclimate temperature reconstructions argue that the 17th, 18th and 19th centuries were relatively cold ("Little Ice Age"), but that roughly 1,000 years ago, there was a relatively warm period ("Medieval Warm Period").

There is, of course, considerable debate over how the Current Warm Period compares to the Medieval Warm Period. The IPCC argues that the Current Warm Period is "likely" warmer. But, notice that even in the IPCC's claim that I quoted above, they only assign medium confidence and the term "likely" (66-100% chance according to the IPCC "likelihood scale") to their claim that "1983-2012 was likely the warmest 30-year period of the last 1400 years" (and that they confine this statement to the Northern Hemisphere).

In other words, the IPCC have actually left it as fairly plausible (up to 34% chance according to the IPCC) that it was similarly warm during the Medieval Warm Period. So, choosing "the past 500 years" as a benchmark (i.e., excluding the Medieval Warm Period) might seem like cherry picking.

Even if you had stuck to the shorter instrumental period (i.e., the last 150 years or so), there is debate over the relative warmth of the recent warm period to the early 20th century warm period. In Soon et al. (2015), we argue that the current land surface temperature datasets have failed to satisfactorily correct for non-climatic biases, including urbanization bias. We developed an alternative estimate of Northern Hemisphere temperature trends since 1881 using mostly rural stations. This new estimate suggests that, while temperatures increased from the 1970s to 2000s, they cooled from the 1940s to 1970s, and temperatures in the 1940s were comparable to present. This could contradict your claim (which you attributed to the IPCC) that the most recent 50 years were the hottest over the past 500 years.

More recently, in Soon et al. (2018), we reviewed the more relevant debate over how the current warm period compares to the early 20th century warm period for China. We showed that there

is considerable debate over how warm the earlier warm period is relative to the current warm period. Indeed, this debate is ongoing, e.g., see Li and Yang (2019) and our reply (Soon et al., 2019). However, hopefully this should illustrate how you need to be very careful in making these sorts of generic statements, when you are referring to the IPCC!

d) Moreover, it is unclear, why you are making this claim anyway. Later (lines 6970), you add, "In the context of global warming, the feedback of the snow cover in China on climate change is unknown.". I appreciate that it has become "fashionable" to include a reference to "global warming" or "global climate change" in any paper looking at climate trends. However, if you really want to frame your results in the context of global warming (as opposed to framing your paper as a study of regional climate change), then you should be more rigorous, and show the full context. For instance, the IPCC claim on p2 of the Summary for Policymakers that, "Warming of the climate system is unequivocal, and since the 1950s, many of the observed changes are unprecedented over decades to millennia. The atmosphere and ocean have warmed, the amounts of snow and ice have diminished, sea level has risen, and the concentrations of greenhouse gases have increased (see Figures SPM.1, SPM.2, SPM.3 and SPM.4). 2.2, 2.4, 3.2, 3.7, 4.2–4.7, 5.2, 5.3, 5.5–5.6, 6.2, 13.2" (emphasis added in bold). Then, on p7, they claim that, "Over the last two decades, [...] Northern Hemisphere spring snow cover [has] continued to decrease in extent (high confidence) (see Figure SPM.3). 4.2–4.7". Yet, this paper shows that, for China, the amounts of (annual) snow cover have generally increased. You are not the first to notice this difference for China (e.g., Wei and Dong, 2015; Wang et al., 2017; Chen et al., 2016 as mentioned above). And, in Connolly et al. (2019), we showed that the IPCC's "spring snow cover" claim was not representative of the overall snow cover trends, e.g., winter and autumn snow cover seems to have generally increased over the same period. With that in mind, I recommend you go with one of two options:

Option 1: Explain how your results are somewhat at odds with the IPCC AR5's claims and discuss the full context. or Option 2: Drop the references to "global warming" and stick to describing the trends for China.

**Author's Response:** After your detailed explanation of the IPCC report, we found that the understanding of 'global warming' for us is so superficial. We maybe too superstitious the report for IPCC. Thank you very much for your carefully explanation, we will carefully to quote

the reference in the future. We adopted your second option, and sticking to describe the trends for China. Thank you.

**Author's changes in manuscript:** The research period is extended from 1951 to 2018.

**2     Comments from Referees:**

2.) Questions on the datasets

a) Do you know if the CMA are planning on updating their snow depth dataset to include more recent years? If not, could you include a comment on why it has not been updated since 2013?

b) On lines 84-87, you say, "To ensure the reasonableness of statistical analysis, we must ensure that the station dataset used for statistical analysis were longer than 10 years. Therefore, the stations with less than 10 years of records were omitted from the analysis.". Can you elaborate on how many stations you had in total and how many were omitted after these steps?

c) Will your results be available as Supplementary Information? If so, this would be great. Ideally, as well as the time series you constructed, I would also like to see the individual station results available as Supplementary Information. But, if that is not possible (since it is a CMA dataset), it would be helpful if you could at least provide some indication of the numbers (and possibly locations) of the stations available for each year.

d) On line 79, you say that the dataset begins on 1 January 1951, but then on line 90, you say, "...since the SD measurements began in October 1951". Which date is it?

**Author's Response:** First, we updated the data to 2018 through our effort. Second, the proportion of length records in years for each stations during the period of 1951 to 2018 are calculated and mapped in Fig 1 based on your suggestion. There 67 stations which the snow records less than 10 years were abandoned in this study based on data quality control. More details on how the snow the snow data handled is described in section 2.1. Third, this dataset is not yet publicly available, as you know, we only have rights use it, but can't publish the data. We are sorry about this, hope you can understand. Fourth, there is confusion in the description of the start date of the dataset, and we have modified it. The hydrological year which spanned from July 1 of the current year to June 31 of the subsequent year is used when analyze the snow variations in this study, thus, the data period used between July 1, 1951 and June 30, 2018. Thank you again for the great suggestion.

**Author's changes in manuscript:**

[Figure]

**Figure 1.** Geographical locations and the proportion of length records in years of each meteorological stations in China mainland.

**3   Comments from Referees:**

3.) Results

a) Should the title of Section 3 be "Results" instead of "Result"?

b) On lines 206-207, you say, "The national $SD_{average}$, $SD_{overall}$ and $SD_{max}$ showed increasing trends from 1952 to 2012 under the context of global warming". What do you mean by this? First, as explained above, I would reconsider whether or not you want to frame this paper in the context of global warming. Also, do you mean "anthropogenic global warming", i.e., an expected global warming trend arising from increasing greenhouse gas concentrations? If so, then as we discussed in Connolly et al. (2019), the CMIP5 hindcasts (which do indeed attribute most of the global temperature trends since the 1950s to anthropogenic greenhouse gas emissions) predict that Northern Hemisphere snow cover should have been decreasing, and not increasing as you find. In other words, your results are actually somewhat problematic "in the context of (anthropogenic) global warming".

c) But, instead of focusing on global trends, why not frame these results simply in the context of regional climate trends for China? How do your results compare with the equivalent

temperature and precipitation trends for China?

In this context, we provided a review of Chinese temperature trends in Soon et al. (2018). You might also find the ensuing discussion, i.e., Li and Yang (2019) and Soon et al. (2019) relevant.

d) As mentioned in the General Comments section, you should include at least some comparison with the Rutgers satellite-derived dataset (for example). Ideally, I would calculate the regional trends for China from their gridded dataset. But, at the very least, your results should be compared to the overall Northern Hemisphere time series.

e) As also mentioned in the General Comments section, you should compare how your results compare to the CMIP5 and/or CMIP6 hindcasts of snow cover for China. See Connolly et al. (2019) for a systematic comparison of Northern Hemisphere snow cover according to observations vs. all available CMIP5 model runs.

f) Have you looked at the differences between seasons? As we discussed in Connolly et al. (2019), the trends for Northern Hemisphere are quite different for each season. Depending on how you carried out your analysis, this might be beyond the scope of your paper, but you should suggest it as a possibility for future work if you think that this could be done using your dataset.

g) To be honest, most of the statistical analysis in Sections 3.4 and 3.5 seems unnecessary to me and doesn't really add much insight (in my opinion). Personally, I would remove these two sections. However, perhaps some readers might find it of interest, so I will leave it up to you whether you think these sections are needed or not.

**Author's Response:** The 'Results' was revised as 'Result' based on your suggestion. And the references about "global warming" were dropped, and sticking to describe the snow variation trends for China. A comparison between the results with the satellite observations as well as CMIP5 model were conducted in Discussion section. However, because the point-based data can't provide too much information like satellite data, thus, the seasonal changes were not analyzed. Based on our results, many of the overall trends are quite modest, which make the wavelet analysis superfluous, and also the structural equation modeling results as you concerned. So we delete the two parts results. Thank you for your suggestions.

4.    Comments from Referees:

4.) Conclusion

Your Section 5 currently reads more like a "summary" than a conclusions. It is ok to include a

brief summary as part of the conclusions, but you should also provide some concluding remarks and/or some recommendations for further research.

**Author's Response:** We have modified the conclusion on the basis of your suggestion. Thanks a lot.

**Author's changes in manuscript:**

'The variation and distribution of snow cover have always been a popular research topic. In this study, we use regression analysis, M-K analysis to analyze a SD dataset provided by the CMA from 1953 to 2018 across China. The SD related to geographical zonality obviously. Latitude and altitude are two key roles in snow distributions. Other main conclusions are condensed as follows:

(1) Snow cover in China is distributed mainly in Northeast China, northern Xinjiang and the Tibetan Plateau. The largest $SD_{overall}$ is in northern Xinjiang (9.3 cm), and the highest number of SCDs is in Northeast China, reaching 167 days. The largest $SD_{max}$(55.3 cm), the longest SDDs, the earliest SOD, and the latest SED are in the Tibetan Plateau.

(2) In the context of the reduction in SDDs, the overall snow depth including $SD_{overall}$, $SD_{max}$, showed increasing trend in most of stations, especially in Northeast and Northern Xinjiang during the period of 1951 to 2018 in China. The SCDs are also showed increasing trend in these two regions. Among the three stable snow cover regions, the Tibetan Plateau showed an opposite variation trend compared to other regions in China. The $SD_{overall}$, $SD_{max}$ and SCDs all exhibited a weak decreasing trend.

(3) The three stable snow cover regions in China showed that the trend of SDDs becoming shorten, which is caused by the delay of the SOD and the advance of the SED, and the advanced SED contributes more to the reduction of SDDs, while in the Tibetan Plateau, the main reason for the shortened SDDs is the delay of the SOD.

In summarize, under the condition that there is no obvious change in snow area in China, the SCDs in high latitudes is increasing, the SOD is delayed, and the SED advanced. However, the SD shows an increasing trend, especially in northeast China, which is inconsistent with the observation results of the whole northern hemisphere, and the causes of this different change pattern need to be worthy of further studied. Warming and precipitation pattern heterogeneity may cause the difference as the author's assumption, the reasons should be further reveal in

which combine the meteorological data in future.'

**5.    Comments from Referees:**

Technical corrections

The written English is not great with some grammatical errors scattered throughout the manuscript. I suggest using the "Check spelling and grammar" option of Microsoft Word or some other similar word processor, and fixing any of the underlined errors that will appear.

Author's Response: The editorial changes for language usage throughout were made by a native English scientific editor in this version. Thank you again for the time you paid on review our manuscript.

**New references updated:**

Brown, R. D., and Robinson, D. A.: Northern Hemisphere spring snow cover variability and change over 1922–2010 including an assessment of uncertainty. Cryosphere, 5, 219–229, doi:10.5194/tc-5-219-2011, 2011

Chen, X., Liang, S., Cao, Y., and He, T.: Distribution, attribution, and radiative forcing of snow cover changes over China from 1982 to 2013. Clim.Change.137, 363–77, doi:10.1007/s10584-016-1688-z, 2016.

Choi, G., Robinson, D., and Kang, S.: Changing Northern Hemisphere snow seasons. J.Clim. 23, 5305–5310, doi:10.1175/2010JCLI3644.1, 2010.

Connolly, R., Connolly, M., Soon, W., Legates, D. R., Cionco, R. G., Velasco Herrera, V. M.: Northern Hemisphere snow-cover trends (1967-2018): A comparison between climate models and observations. Geosci., 9, 135, doi:10.3390/geosciences9030135, 2019.

Derksen, C., and Brown, R.: Spring snow cover extent reductions in the 2008-2012 period exceeding climate model projections. Geophys. Res. Lett., 39, L19504, doi: 10.1029/2012GL053387, 2012.

Euskirchen, E. S., McGuire, A. D., Chapin, F.S.: Energy feedbacks of northern high-latitude ecosystems to the climate system due to reduced snow cover during 20th century warming. Glob. Chang. Biol. 13, 2425–2438, doi:10.1111/j.1365-2486.2007.01450.x, 2007.

Hall, D.K., Riggs, G.A., Salomonson, V.V., DiGirolamo, N.E., Bayr, K.J. MODIS snow-cover products. Remote Sens. Environ. 83, 181–194, doi: 10.1016/S0034-4257(02)00095-0, 2002.

Hori, M., Sugiura, K., Kobayashi, K., Aoki, T., Tanikawa, T., Kuchiki, K., Niwano, M., Enomoto, H.: A 38-year (1978–2015) Northern Hemisphere daily snow cover extent product derived using consistent objective criteria from satellite-borne optical sensors, Remote. Sens. Environ., 191, 402-418, 2017.

Peng, S., Piao, S., Ciais, P., Friedlingstein, P., Zhou, L., Wang, T.: Change in snow phenology and its potential feedback to temperature in the Northern Hemisphere over the last three decades, Environ. Res. Letters., 8, 014008, 2013.

Percival, D. B., Walden, A. T.: Wavelet Methods for Time Series Analysis (Cambridge Series in

Statistical and Probabilistic Mathematics), UK: Cambridge Univ. Press, 2000.

Wei, Z., and Dong, W.: Assessment of Simulations of Snow Depth in the Qinghai-Tibetan Plateau Using CMIP5 Multi-Models, Arctic Antarct. & Alp. Res., 47, 611-525, 2015.

---

## Author Comment (AC3) · 28 Nov 2019

Dear Prof. Florent Dominé, Enclosed is the revised manuscript. On behalf of all authors, we appreciate your help and the time you paid on our manuscript. We also appreciate the two reviewers for their valuable and constructive comments and suggestions. In the revised manuscript, we have revised our manuscript and answered the questions on a point-by-point basis base on your and reviewer's comments. The editorial changes for language usage throughout were made by a native English scientific editor in this version. The revised manuscript has made the following changes: 1) The snow data from 2014 to 2019 is collected through our effects, which extend the period from 1951 to 2018. We recalculated and rewrote the results and discussion sections in the revised manuscript. 2) Cause we don't have the supporting data like

temperature and precipitation to prove our results in this manuscript. So the reference involved in "global warming" were dropped, and stick to describe the snow cover trends for China only based on professor Connolly's comments. 3) The snow variations trends are compared with the results from remote sensing observations as well as the simulation results from CMIP5 model. We rewrote the discussion section, and focus on the consistent with other current results to further prove the findings in this study.

Best regards,

Xiaodong Huang